



# Review Article: Earth observations of Melt Ponds on Sea Ice

Sara Aparício[1]

[1]CENSE - Center for Environmental and Sustainability Research, NOVA School of Science and Technology, NOVA University, Lisbon, Portugal

*Correspondence to*: Sara Aparício (s.aparicio@campus.fct.unl.pt)

**Abstract.** Melt ponds are pools of open water that form during summer on sea ice surface, playing a key role in the Arctic climate and sea ice energy budget. Due to their lower albedo, melt ponds absorb more radiation than un-ponded ice, spurring further ice melt and enhancing the positive ice-albedo feedback. This feedback is expected to increase as the Arctic ice cover

is moving towards a regime where multiyear ice (MYI) is being replaced by first-year ice (FYI), which presents wider melt pond coverage as well as more energy absorption with profound consequences from an energy balance perspective. Nevertheless, the lack of knowledge or inclusion of melt pond fraction (MPF) on global climate and sea ice models, is pointed as their main source of uncertainty and disparity of predictions results. This, along with the recent conclusions on the potential of MPF for enhancing the forecasting ability of summer sea ice extent, underscores the importance of accurately obtaining

large and spatiotemporal scale of MPF, across the Arctic. However, observations of melt ponds are far from adequate for the Arctic ocean and for both MYI and FYI,  and on a large scale this is only  possible through satellite-based Earth observations (EO). This paper provides an overview of efforts in EO remote sensing studies of melt ponds, for both optical sensors and radar-based approaches. The main algorithms used for melt pond identification and the different methods for MPF retrievals are outlined, ranging from the early traditional techniques to the increasingly prevalent use of Artificial intelligence (AI),

namely machine and deep learning. The current large-scale optical-based pan-Arctic MPF datasets are intercompared along with the main advantages and disadvantages of various optical and radar data-based methods for MPF retrievals. The potential of radar, namely Synthetic Aperture Radar (SAR) technical abilities to the enhancement of reliability is analysed, since optical approaches, despite being more used, are hampered by cloud cover, spectral representativeness and resolution. Finally, current gaps in melt pond knowledge and MPF retrievals are discussed and summarised leading to the outline of further directions of

research development.

## 1 Introduction

Melt ponds are one of the most distinctive features of the Arctic sea ice surface playing a pivotal role in the Arctic energy budget (Curry et al., 1995; Eicken et al., 2004). These pools of open water form during summer and spring, as a consequence of melted snow and ice being a persistent sea ice characteristic during this period (Fetterer and Untersteiner, 1998), which can

also be found on ice shelves and glacial ice (Fig.1). They occupy a large fraction of the Arctic sea ice surface (Fetterer and





Untersteiner, 1998), contributing to its energy absorption and melting processes (Polashenski et al., 2012). Although melting sea ice is a seasonal process, the melt season is lengthening (e.g. Perovich et al., 2008; Markus et al., 2009; Rösel and Kaleschke, 2012; Pistone et al., 2014) and becoming significantly stronger than before (IPCC, 2014), raising a particular interest in melt ponds' dynamics and impact in the global climate context.

The Arctic region has experienced amplified warming in recent decades (ACIA 2004; IPCC, 2014), which is linked to the decline of the volume, thickness and extension of Arctic sea ice (Stroeve and Notz, 2018) since the beginning of the satellite record in 1979 (Stroeve et al., 2012). One of the fundamental challenges of climate science is to develop more rigorous representation of sea ice in climate models and to incorporate small scale processes and structures (Barjatia et al., 2016), such as the complex evolving mosaic of ice, open water and melt ponds that occurs during the melting season. As such, it is

important to understand changes in sea ice properties, in particular the coverage of melt pond - since it is the main factor affecting albedo of sea ice (Eicken, 2004), in order to parameterize sea ice in global climate models (Flocco et al., 2012).

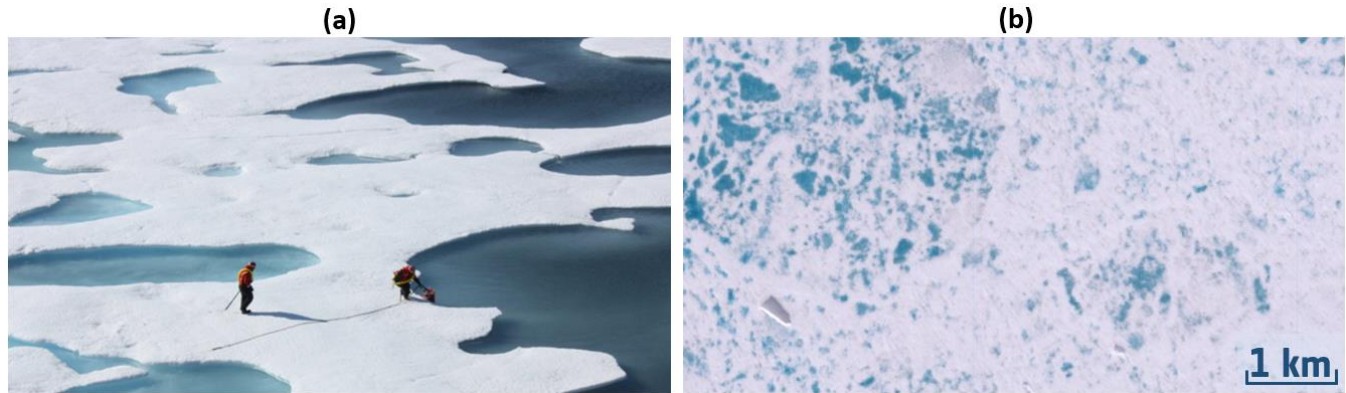

**Figure 1: Melt ponds on Arctic sea ice: (a) Photograph by Kathryn Hansen in July 2012 | NASA Goddard Space Flight Center used under CC BY 2.0 and (b) Copernicus Sentinel-2 True colour (RGB) image acquired in June 2016 in coastal East Greenland**

## 2 Melt ponds and sea ice energy budget

Melt ponds affect the radiative balance in the Arctic as they drastically reduce the ice albedo ($\alpha$) (Curry et al., 1995; Eicken et al., 2004). Albedo ($\alpha$) (or spectral reflectance), that can be defined as the ratio of solar energy reflected upward over energy incident upon the same surface (Hanesiak et al., 2001)). The darker colour from the accumulation of meltwater on the surface

of the ice lowers the albedo, ranging from ~0.8 down to 0.2-0.4 (Perovich et al., 2002), enhancing the flux of absorbed solar shortwave radiation. This contributes to speeding up the melting process, amplifying the positive ice-albedo feedback mechanism (Curry et al., 1995; Eicken et al., 2004; Schröder et al., 2014; Liu et al., 2015; Webster et al., 2015; Zege et al., 2015). In fact, Fetterer and Untersteiner (1998) estimated that the melting speed of pond-covered ice was between 2 and 3 times faster than of bare ice. Ice albedo's temporal and spatial variation is closely related to the global climate change and





regional weather system due to the above mentioned feedback mechanism (Henderson-Sellers and Wilson, 1983), consisting

in one of the most crucial parameters governing the climate processes in the Arctic (Dickinson et al., 1983; Qu et al., 2015; Zege et al., 2015).

The summer albedo of melting Arctic sea ice is strongly affected by the fraction of the surface which is covered by melt ponds, commonly referred as melt pond fraction (MPF) (Eicken et al., 2004; Polashenski et al., 2012) and also by their surface

topography (Eicken et al., 2004). They can present light-blue colour when they just form, or darker tones if they are deeper– having thus a wide range of albedo values. This means that the characteristics of the ponds are important in addition to their coverage (Sankelo et al., 2010; Polashenski et al., 2012).

Besides the ice-albedo feedback, melt ponds also have other effects on Arctic climate, including their impact on redirecting freshwater between ocean and ice systems, delaying the re-stratification of the ocean surface layer, and contributing to the heat

and momentum fluxes through their edges (Sterlin et al., 2020). Finally, melt ponds also affect the salt and heat budget of the ocean surface layer (Landy et al., 2015) partly contributing to the observed ice loss, making this phenomenon as important and inevitable as the dramatic decay of current Arctic sea ice (Flocco et al., 2012). Besides their climate importance, melt pond evolution has been also linked to biological impacts, as increased penetration of light to the upper ocean through thin first-year ice, may lead to massive under-ice phytoplankton blooms (Arrigo et al., 2012).


## 2.1 Melt ponds' radiative properties

Earlier studies on geophysical and radiative properties of melt ponds (e.g. Grenfell and Maykut,1977), have identified parameters such as melt pond fraction, runoff fraction, solar radiation absorbed by the water, reflection and transmissivity, which contributed to the current understanding of summer sea ice. The factors governing the optical properties of the ice are

sensitive to the temperature of the material (which undergoes significant variation with season); the state of the upper surface; the age of the ice (Grenfell and Maykut, 1977) and the shape and size of the ice grains- which are much greater than the wavelength of the visible light (Kokhanovsky and Zege, 2004). Optical properties of ice change mostly during the summer melt season, as result of snow decay, bare ice exposure and formation of meltwater puddles. Since polar snow can undergo such large changes in internal structure, its optical properties span a much broader range of values than do those of the sea ice

(Grenfell and Maykut, 1977). The most studied, and most used, optical property of sea ice is the albedo, since it plays a fundamental role in the understanding of the Arctic climate processes (e.g. Perovich et., 2002). It is normally used specifically for reflected visible and infrared light - which can be retrieved by optical sensors. Although surface albedo studies have been conducted and extensively summarised during the 60s, the first studies dedicated to sea ice were conducted by Grenfell and Maykut in 1977, by measuring spectral albedo of various sea ice and melt ponds from 400 to 1000 nm. Since water is mostly

transparent from 400 to 500 nm, and scattering is dominant within the ice, the spectral albedo of melt ponds is relatively high at these wavelengths. After 500 nm, the increase of absorption by water drastically decreases the pond spectral albedo as shown in Figure 2.





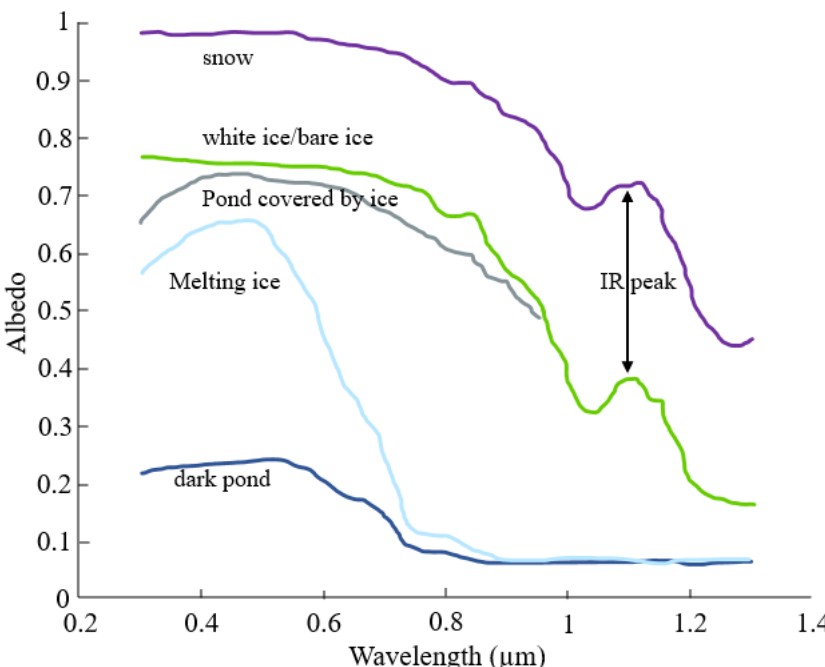

**Figure 2: Spectral albedo of different surfaces. Adapted from Grenfell and Maykut (1977) and Lu et al. (2018)**


The white ice, or bare ice has a substantial surface scattering layer, providing a stable high reflectance varying between 0.75-0.8 in the blue-green region of the spectrum (at 450-500 nm) (Perovich et al., 2002). The increase of snow grain size (which occurs with increment of liquid water content in snow) is responsible for reducing albedo (Warren, 1982). The broadband albedo of ponds (i.e. albedo within a certain wavelength range) can vary between 0.1 and 0.7 (and between 0.1 and 0.4 at red

wavelengths), depending principally on the underlying ice thickness (Istomina et al., 2016; Lu et al., 2018). More specifically, values range from 0.2 over dark melt ponds to 0.4 for light melt ponds whereas the typical values for melting ice range between 0.6-0.7 (Perovich et al., 2002). Ice and water absorb a significant amount of light in the infrared (IR) range, contrastingly to the visible range. For this reason, melt pond optical response in the IR is restricted to the Fresnel reflection by the pond surface (Malinka et al., 2018). For wavelengths longer than 0.9μm the melt pond is restricted by the Fresnel reflection while snow and

white ice demonstrate a local maximum at 1.1 μm (Malinka et al., 2018). In contrast to the visible range, ice and water absorb a significant amount of light in the IR part of the spectrum. The spectral signature (i.e. reflectance as a function of wavelength) refers thus to the relationship between the wavelength (or frequency) of electromagnetic (EM) radiation and the reflectance of the surface. This signature is affected by several things (such as material composition), and can be used to infer information about the surface such as its composition (e.g. water, ice, etc.). Each material has a unique signature, and for that reason it can

be used for material classification (e.g. Huck et al., 2007). As melt ponds form and evolve there are large changes in their





spectral signatures at visible and near-infrared wavelengths (Tschudi et al., 2008). Figure 2 shows also an example of the albedo spectral dependence for white ice, snow and two types of melt ponds - light and dark.

The main factor determining the albedo of a pond as a whole is its albedo bottom (Lu et al., 2016), but only recently  efforts
have been made on trying to calculate it. Malinka et al. (2017) developed for the first time a way to measure it, considering the inherent optical properties of sea ice. The albedo of a melt pond bottom is itself determined by the inherent optical properties as well as the underlying ice thickness. The shallowness of melt ponds on Arctic sea ice is a challenge for classical depth retrieval, since pond albedo in the visible wavelength region of the EM spectrum is primarily defined by pond bottom characteristics (Lu et al., 2016), making RGB images unsuitable to map melt pond bathymetry (Lu et al., 2018). The spectra
of melt ponds albedos is also affected by the illumination condition and background albedo, i.e. black sky results in higher albedo comparatively to white sky. Correct processing of the reflection measurements requires the correct modelling of the illumination conditions. This is of especial importance on Arctic measurements – due to the low sun and bright surface (Malinka et al., 2018).

The colour of melt ponds varies from light bluish to dark as it can be seen in Figure 1,  largely depending on the age of the pond and the properties of underlying ice (Lu et al., 2018; Buckley et al., 2020). These colour variation can be easily examined during field investigations. The first quantitative measurements of melt-pond colour were performed in the central Arctic in 2012 (Istomina et a., 2016). More recently, Lu et al. (2018) explored how colour-based methods could be used to retrieve information on pond depths and underlying ice thickness. They concluded that both depths and underlying ice thickness have
an important impact on pond colour. The green and blue intensities increase only with increasing ice thickness, except for very thick ice (>4m), but the red intensity increases mostly with increasing ice thickness  for thin ice (<1.5m) and increasing depth for thick ice (>1.5m), similarly to melt-pond albedo. The reproduced pond colour gradually changes from dark blue to bright blue with increasing ice thickness (Lu et al., 2018). Buckley et al. (2020) also concluded that pond colour is also dependent on the ice type on which ponds form. They can present light-blue colour when they just form, or darker tones if they are deeper.
This means that the characteristics of the ponds are important in addition to their coverage (Polashenski et al., 2012, Sankelo et al., 2010).

## 2.2 Formation and evolution of melt ponds across different ice types

The evolution and distribution of melt ponds are functions of snow thickness, snow distribution, and sea ice type (Fetterer and
Untersteiner, 1998; Eicken et al., 2004; Scharien and Yackel, 2005). Melt ponds start to form in the Arctic at the end of May, covering a considerable portion of the sea ice surface by mid-June (Polashenski et al., 2012; Rösel and Kaleschke, 2012). Throughout June and July, the ponds widen and deepen and begin to refreeze in early September or end of August (Fetterer




and Untersteiner, 1998). The seasonal pond evolution is generally divided into four stages: a) melt onset, b) drainage , c) melt evolution and d) freeze up as detailed by Eicken et al. (2002) and Lei et al. (2016) and summarised in Figure 3.

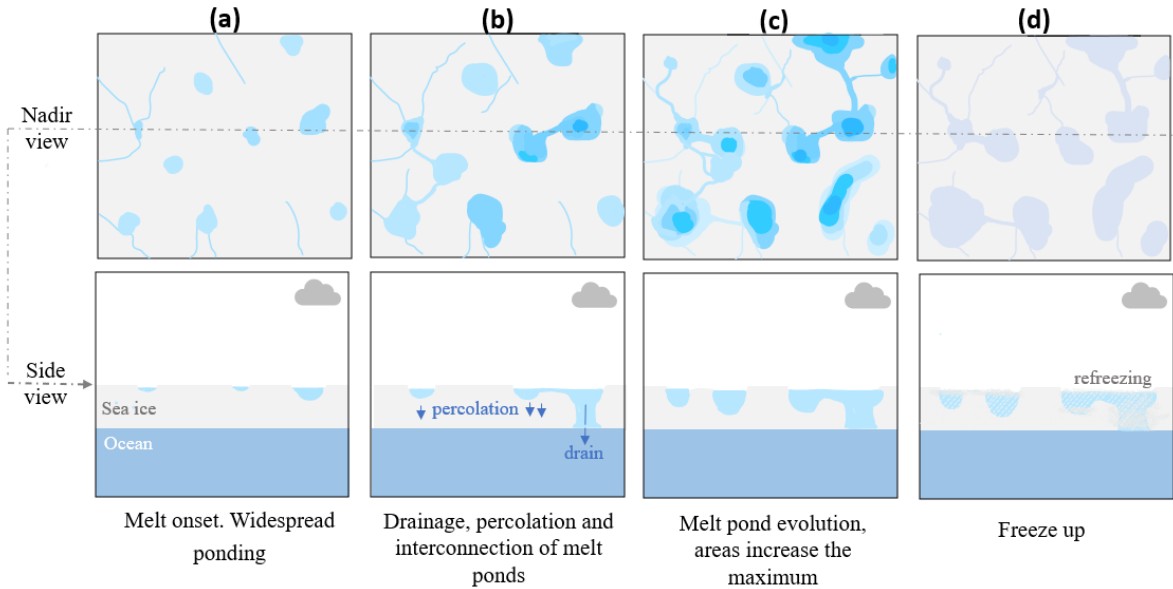


**Figure 3: Main characteristics of the 4 stages of melt pond evolution following Eicken et al. (2002); Polashenski et al., (2012); Lei et al. (2016) and Li et al., (2020): a) Mel onset and widespread ponding, (b) Drainage, percolation and interconnection of melt ponds, (c) Melt pond continue to evolve and area covered increases to the maximum and (d) Freeze up**

During the first stage meltwater accumulates in cracks and depressions (Eicken et al., 2002)). The albedo drops to ~0.6 in the early stage due to the snow melt, decreasing further down to ~0.32 associated with the formation of the melt ponds (Perovich and Polashenski, 2012). During the second stage with the increase of ice temperature, the permeability of ice increases resulting in percolation through drainage through macroscopic flaws (Eicken et al., 2002; Polashenski et al., 2012), causing a slight increase of the albedo to ~0.54 (Perovich and Polashenski, 2012). Furthermore, in this stage, the horizontal transport of
meltwater results in the interconnection between streams and ponds. Also during this phase many ponds are created expanding the coverage of melt ponds. In the third stage, the melt ponds are widely distributed, reaching the seasonal maximum (Polashenski et al., 2012). The albedo of mature ponds is generally below 0.3 or even 0.2 in the case of seasonal ice (Perovich and Polashenski, 2012). Finally, the fourth stage consists in refreezing although it is not restricted to the end of the season. This can in fact stop meltwater inflow and cause a skim of ice to form over the ponds. This, in addition to snowfall, can
temporarily remove the albedo effect of the ponds (Grenfell and Perovich, 2004), as observed by Anhaus et al. (2021). Since mid-August is associated with the freeze-up, the albedo gradually increases. The evolution of melt pond coverage during the decay period or last stage, presents a relatively asymmetrical behaviour compared to the growth period (Ding et al., 2019) which is associated with the surface changes and snowfall (Rösel and Kaleschke, 2011; Rösel et al., 2012; Istomina et al., 2015).




Melt pond formation and progression presents considerable differences between first-year ice (FYI) and multiyear ice (MYI)(Webster et al., 2015; Buckley et al., 2020; Li et al., 2020). FYI is ice that grows in fall and winter, but does not survive the spring and summer months (i.e., it melts away). This type of ice has thus no more than one year growth and typically has a thickness ranging from 0.3 to 2 m. On the other hand, MYI is ice that has survived at least one melting season, and for this

reason is generally thicker than first-year ice. Typical thickness of FYI and MYI are 1.5 and 3m, respectively (Zhang et al., 2019). Field experiments have shown that melt ponds are wider and shallower on FYI, and smaller but deeper on MYI (Eicken et al., 2004; Polashenski et al., 2012; Wright et al., 2020), which has been supported by satellite based observations (e.g. Ding et al., 2019). The ice-ponded coverage is highly variable going from zero percent in mid-June to as high as 80 or 90% (Fetterer and Untersteiner, 1998), and presents some differences between the ice type. FYI has a higher fractional pond coverage,

reaching up to 75% on landfast ice (Polashenski et al., 2012; Grenfell and Perovich, 2014; Hanesiak et al., 2001), comparatively to multi-year ice, with typical values ranging between 20 and 40% (Polashenski et al., 2012; Rösel et al., 2012). Also the albedo of melt ponds present differences between the ice type where it occurs, lower than 0.2 in FYI and approximately 0.35 in MYI (Perovich et al., 2007).

Besides wider pond coverage on FYI, melt ponds also develop faster on this type of ice, in comparison to MYI evolution, which is much slower (Ding et al., 2019; Li et al., 2020). One of the reasons why FYI tends to experience greater pond coverage than MYI (Wright et al., 2020; Buckley et al., 2020) is linked to the topography of both ice types. When surface melt begins, melt water is unconstrained by topography in FYI (since it tends to form in flat undeformed pans) and spreads to cover a large fraction of the surface. The ponds spread laterally across the level across the level first year ice surface, and can melt out the

thin ice by the end of the season (Fetterer and Untersteiner, 1998). On MYI, however, meltwater is then contained by prior years' melt-formed topography into well-defined pools, due to the more complex surface topography from ice that has survived prior melt seasons (Wright et al., 2020; Buckley et al., 2020). The topography of MYI inhibits the lateral spread of melt ponds across the surface (Eicken et al., 2004) and instead, drainage channels form, connecting ponds as shown in Figure 1. In addition, FYI has more brine pockets and less air bubbles comparatively to MYI, resulting in a decreased scattering causing a

more absorption of radiation contributing to more melting in this ice type (Scott and Feltham, 2010).

In the late 1970s, two-thirds of the Arctic Basin was occupied by MYI, but in comparison, FYI now covers over two-thirds of the area (Kwok, 2018; Zhang et al., 2019). In other words, as sea ice extent and thickness decreases, thick multiyear sea ice MYI has been replaced by thinner first-year ice (Kwok, 2018). This change of proportion has influenced the weather and

climate through different radiation and dynamic properties (Perovich et al., 2002). The replenishment of MYI with FYI increases the heat exchange between the atmosphere and the ocean because FYI is thinner and easier to deform (Stroeve et al., 2012). This would contribute to the positive ice-albedo feedback earlier described, since higher pond fraction would lower albedo of FYI, reinforcing the transition to a younger ice. In addition, the prevailing sea ice type has been changing from MYI





to FYI in recent decades (Comiso, 2012), meaning that the topography is also subject to change from rough to uniform, flatter

surface, which is expected to result in the increase of melt pond coverage (Scott and Feltham, 2010). As ice transitions into a
young and thinner ice cover, it is expected to see changes in melt pond coverage, albedo, depth and volume. In situ
measurements with a ROV under sea ice showed that FYI extensively covered by melt ponds, allow three times as much light
to pass through than that of MYI. In addition, it also absorbs approximately 50% more solar radiation (Nicolaus et al., 2013).

As the melt pond fraction is closely related to the relief of the sea ice (Polashenski et al., 2012), the maximum melt pond
fraction (MPF) is equally expected to increase (Rösel et al., 2012; Malinka et al., 2018), as it has been observed in years of
extreme sea ice extension loss (Rösel et al., 2012). Nevertheless, a decreasing trend of overall MPF was evident when analysing
MPF dataset covering periods of many years (Peng et al., 2022) which can be due to the decrease of sea ice extent. Recent
studies covering the melting season (May-August) from 2000 to 2019, show that as a results of advances in Arctic melt onset

and the extension of the melting period, the MPF as a whole shows a significant upward trend in addition to a significant
increase of MPF in multiyear ice (Feng et al., 2022). As the proportion of MYI and FYI changes, so does the melt pond type,
coverage and contribution to the increase of ice-albedo feedback, since light transmittance and energy absorption will increase
in the future (Scott and Feltham, 2010; Nicolaus et al., 2013) as represented in Fig.4. Also, according to recent observations,
the melt onset is shifting earlier and the whole melt season is getting longer (Perovich et al., 2008; Markus et al., 2009; Rösel

and Kaleschke, 2012; Pistone et al., 2014).

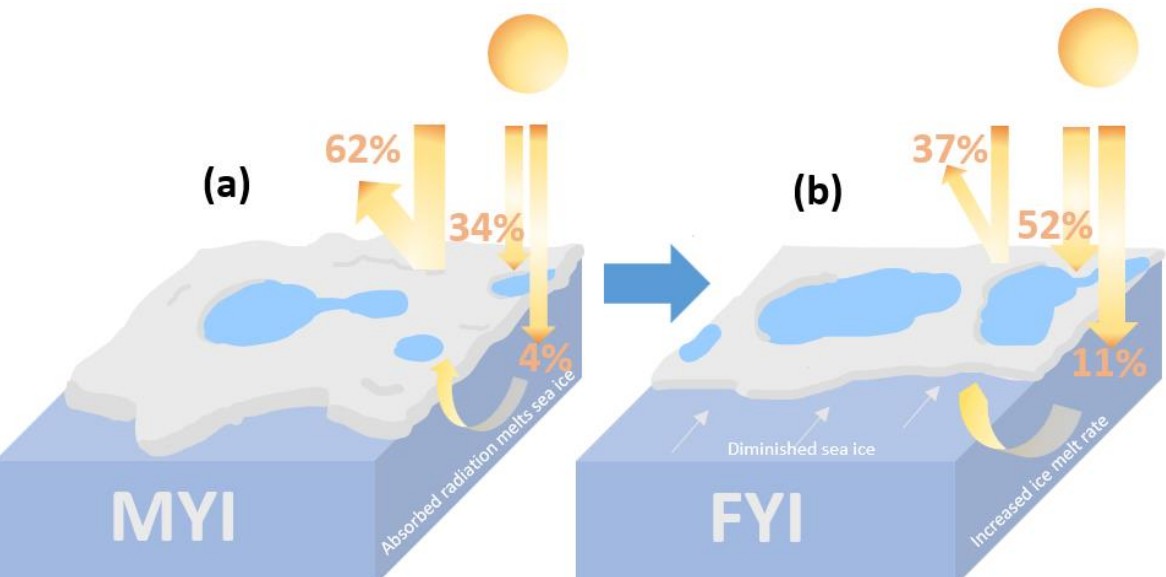

**Figure 4: Evolution of reflection and absorption of solar radiation - and consequent feedback mechanism following the trends of and trends on replenishment of (a) MYI with a thicker irregular surface to a (b) more flatten, thinner and ponded FYI. Adapted from Nicolaus et al. (2013).**




## 3. Remote sensing studies of melt ponds

The majority of the current understanding of melt ponds comes from in situ observation, since it can provide accurate and precise measurements of melt ponds properties, however, it is costly and restricted by accessibility limitations. Large-scale aerial observations are only achieved with aerial survey and/or satellite remote sensing, and because of that Earth observation
(EO) remote sensing is gaining a more relevant role during the last decades. Current remote sensing methods for melt ponds mapping can be categorised as either airborne or spaceborne. Airborne platforms normally include aerial photography (with digital cameras)(e.g. Perovich et al., 2002) normally onboard drones, helicopters or unpiloted aerial vehicles (UAVs) (Tschudi et al., 2008; Fetterer et al., 2008), such as the Surface Heat Budget of the Arctic Ocean (SHEBA) program. SHEBA was based on a drifting ice camp in the Beaufort Sea between October 1997 and October 1998. This program contributed to a better
understanding of melt ponds' albedo, fraction, and size distribution (Perovich et al., 2002), as well as input for MPF and melt ponds depth parameterization in modelling efforts (e.g., Curry et al., 2001). Although airborne studies have demonstrated the ability to extract melt pond parameters from imagery, most are spatially and temporally limited. For example, Miao et al. (2015) classified images collected in the Arctic Pacific sector over a single summer, whereas Tschudi et al. (2001) and Perovich et al. (2002) analysed data collected during the last year of the campaign. Nearly a decade later, in the summers of 2016 and
2017, the NASA Operation IceBridge (OIB), obtained to that date the most widespread airborne survey of summer sea ice conditions. The NASA 524 HU-25C Guardian aircraft carried multiple instruments on board (radar, laser altimeter and a digital camera), and covered both MYI and FYI in the Beaufort Sea (in 2016) and MYI in the Lincoln Sea and central Arctic Ocean (in 2017). Although data from this campaign was used to retrieve melt pond fraction (MPF) (e.g. Wright and Polashenski, 2018; Buckley et al., 2020), it has also been used for independent validation of the development of an algorithm to retrieve
melt pond identification of MPF (e.g. Wang et al., 2020).

The difficulty in deploying in situ instruments in remote locations as the Arctic, as well as the fact that in situ campaigns are cost- and labour-intensive and spatially limited, makes Earth observation (EO) attractive for monitoring sea ice. Satellite-based EO allows for larger scale gathering of imaging of Earth's surface, thus enabling better coverage and information retrieval from polar regions thanks to the satellite mounted payloads. However, even though satellite imagery provides more Arctic
coverage, it frequently sacrifices resolution. High resolution imagery is required to resolve the geophysical features and evolution of melt ponds, which can have an area of 10 m2 at melt onset (Perovich et al., 2002) and expand throughout the melt season. Remote detection of melt pond and their physical and radiative properties, i.e. interaction of electromagnetic waves ranging from UV to IR, initiated with more consistency roughly twenty years, however some studies are dated back to late 90s (e.g. Comiso and Kwok, 1996: Fetterer and Untersteiner, 1998). The first studies done on melt ponds used mostly optical (e.g.
Perovich et al., 2002; Markus et al., 2003; Miao et al., 2015; Webster et al., 2015), but there has been a growing interest in radar due to its capability of penetrating clouds. Some of the earlier studies using radar, used aircraft radar to describe melting processes (e.g. Holt and Digby, 1985).



## 3.1 Passive optical, and active and passive microwave sensors principles

The vast majority of optical sensors are passive, as they collect imagery in the visible or near-visible portion of the electromagnetic spectrum, using the sun's radiation reflected by our planet. Optical satellites capture spectral bands on the visible side of the EM spectrum with some additional capabilities in adjacent wavelengths. When images combine data from at least three different spectral bands, they're referred to as multispectral images. Taking, for instance, the Copernicus Sentinel-2 as an example. This polar orbiting mission, composed of two twin satellites Sentinel-2A and B, launched on 23 June 2015 and March 2017, respectively, carries a multispectral instrument (MSI), which samples 13 spectral bands (Fig.5a). From these 13 bands (443-2190 nm), three are the visible RGB bands (red, green and blue), four bands are in the visible and near infrared (VNIR) and the rest lie in the short wave infrared (SWIR) part of the EM (Sebastianelli et al., 2021) as shown in the pictures (Fig.5b). When viewing the output from just one band, the brightest spots are areas that reflect or emit a lot of that wavelength of light whereas the darker areas reflect or emit little (if any). When combining the actual measurements bands of red, green and blue, represented into the red, green and blue colour channels, the result is a natural or "true colour" image (Fig.5c). One common approach to retrieve information from optical imagery  is through the combination of several bands of the optical sensor, which are hence referred to as "false colour" images (Fig.5c). A false-colour image uses at least one non-visible wavelength, though each band is still represented in red, green or blue, and it has the advantage of revealing aspects of the surface that might not be visible otherwise. Similarly, to "false colour" images, another common practice is the generation of a single band, normally referred to as index, calculated from the computation between two bands, for instance the Normalised Difference Snow Index (NSDI), which is computed the normalisation with green and short wave infrared (SWIR) spectral bands,  mostly used in snow/ice cover mapping applications (Hall and Riggs, 2011). In order to show the features of interest, a common approach is to select the wavelength bands which are most likely to highlight those features. For instance, in the context of melt ponds, near infrared (NIR) light wavelengths are useful for discerning land-water boundaries that are not obvious with visible light, since water absorbs NIR. On the other hand, in respect to shortwave infrared light, water absorbs light in three regions, meaning that the more water there is, the darker the image will appear at higher wavelengths (i.e. ~1400, ~1900 and ~2400 nm). This makes SWIR also useful for applications to distinguish between cloud types, snow and ice which normally appear white in visible light.

For active sensors with longer wavelengths, clouds and precipitation have almost no effect on the propagation of electromagnetic waves through the atmosphere. This provides the opportunity to monitor sea ice regardless of the season, illumination or weather-related phenomena. The radar systems used to observe sea ice are altimeters, scatterometers and Synthetic Aperture Radar (SAR). SAR is a radar imaging technique that measures the return signal from a sequence of pulses emitted from a side looking geometry, perpendicular to the line of flight. Using Copernicus Sentinel-1 as an example (Fig.5d), this polar orbiting mission, also composed by two twin satellites (Sentinel-1A and Sentinel-1B, launched respectively in April 2014 and April 2016), carries on board a SAR operating at C-band (central frequency of 5.404 GHz, wavelength around 5cm) (Sebastianelli et al., 2021). It synthesises a larger aperture which allows for the improvement of spatial resolution in the azimuth





direction. The EM signal is emitted by the antenna, which is reflected or scattered by the surface, and then collected by the antenna (Fig.5e), hence, a single radar image, usually displayed as a grayscale image, in which the intensity of each pixel represents the proportion of microwave backscattered from that area (which depends on a variety of factors). The pixel intensity, or backscatter coefficient ($\sigma\circ$) can range from +5 dB for to -40dB for very dark surfaces (such as calm water surfaces

of a melt pond as shown in Fig.5f) .







**Figure 5: Comparison of data acquisition by optical and radar satellites. (a)** Copernicus Sentinel-2 data acquisition by multispectral instrument (MSI); **(b)** S2 band information with discrimination of band identification, central wavelength (µm) and resolution (m) and **(c)** examples of band combination of a summer scene of a ponded sea ice surface (left: B4-red, B3-green, B2-blue bands; right: B2-blue, B6-red edge and B9-SWIR bands); **(d)** Imaging for a typical strip-mapping synthetic aperture radar imaging system. The antenna's footprint sweeps out a strip parallel to the direction of the satellite's movement. **(e)** Transmission and reception of radar pulse: (left) the radar pulse is transmitted from the antenna to the ground and (right).then it is scattered by the ground targets back to the antenna. **(f)** Image collected from backscattered information.

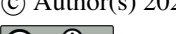



Finally in regards to passive remote sensing, microwave radiometers (which have been used since 1978) observe the thermal
radiation emitted by the surface at microwave frequencies, i.e. on the order of tens of gigahertz. The radiometer which collects
the microwave signals, is calibrated to convert the measured power into a *brightness temperature*, which is a measure of the
amount of thermal radiation emitted by the surface (and in other words, the brightness temperature of a black body surface
emitting the same amount of energy over that frequency interval), enabling the comparison of the radiation emitted by different
surface types at the same frequency. In the case of melt ponds, the microwave-band emissivity of water is much higher than
of ice enabling the use of increases in brightness temperature as an indicator for the presence of melt over snow and ice (Mote
et al., 1993). Most melt detection methods use 19 GHz (e.g. Kern et al., 2016; Johnson et al., 2020) in horizontal polarization
since it is the frequency that has the lowest brightness temperature over dry firn, which maximises the change in brightness
temperature caused by the emergence of liquid water (Johnson et al., 2020). The optical depth of a few millimetres water layer
in a forming melt pond is enough to diminish the microwave signal of the sea ice beneath and the melt pond has a brightness
temperature signal of open water, meaning that the in sea ice concentration (SIC) retrievals the fraction of open water also
includes the melt pond in addition to the open water in cracks and leads, resulting in a major shortcoming of today's suit of
SIC algorithms (Rösel et al., 2012; Kern et al., 2016).

### 3.1.1 Considerations on radar-based signature of melt ponds

The radar parameters affecting melt ponds signature are the resolution, the frequency, incidence angle and polarization (e.g.
Kern et al., 2010; Fors et al., 2017). The frequency of the SAR is one of the key parameters affecting melt ponds' SAR
signature. C-band (5.4 GHz) is less sensitive to small-scale surface roughness than higher frequencies such as X-band (9.6
GHz) (Kern et al., 2010), since the effect of surface roughness depends on radar wavelength. For this reason, X-band has a
strong sensitivity to micro-sale surface roughness due to the high frequency (Fors et al., 2017). Furthermore, the higher the
frequency, the lower the sea ice volume penetration depth, which leads reduced volume scattering from sea ice at higher
frequencies. Another important radar parameter affecting melt ponds' signature is the incidence angle. The greater the
incidence angle, the lower the backscatter intensity ($\sigma°$) of C-band SAR (Scharien et al., 2012). However, the best incidence
angles for melt pond or melt pond fraction retrievals will depend on the method or the band frequency (Fors et al., 2017). The
resolution of the SAR image can also affect melt ponds signature. For instance when ponds are smaller than the SAR resolution,
it could return a signal mixed with sea ice and leads.


Another major parameter affecting melts ponds' radar signature is polarization. Polarization consists in the direction of the
electric field vector of a propagating EM, where the electric vector is transverse to the direction in which the EM energy is
transmitted (Boerner, 2003; Woodhouse, 2006). When a radar emits a pulse (which can be for instance horizontally (H), or
vertically (V)), it picks up varying polarization information when the energy returns to the receiver. SAR has progressed from
conventional single-polarization (single-pol, VV or HH) to quad-polarimetric mode (quad-pol), which includes HH, HV, VH
and VV) (Scharien et al., 2010, 2014a, 2014b). These four possible channel combinations give information about the





polarization properties of the backscatter in addition to single-channel intensity variations (Fors et al., 2017). The channels can be combined into polarimetric SAR features e.g. channel ratios, reducing the dependency of sensor geometry. It is of general consensus that multiple-pol SAR plays a unique role in geophysical remote sensing. Special focus is made to quad-pol SAR,

as it measures the whole scattering matrix for each resolution cell, providing both intensity and phase information (Li et al., 2017), which can lead to the classification of the scattering mechanisms of the sensed targets, with more robustness and physical reliability than sing-pol images. By rationing, the emphasis is on the relative properties of different polarization rather than the absolute intensity of the signal (Woodhouse, 2006). This means that ratios are more indicative of geometric or shape properties of a target, rather than dielectric or local illumination conditions of a target. There is no universal agreement as to

whether the H or V should be used as a denominator, however, the horizontal polarization has been used as the normalising factor. The ratio of vertical to horizontal signal, $r$, relates directly to the expression of backscatter in active sensors such as SAR (or to the reflectivity, for passive sensors). For active sensors, *co-pol ratio* is calculated following Eq.1 :

$$r_{co-pol} = \frac{|S_{VV}|^2}{|S_{HH}|^2} = \frac{P_{VV}}{P_{HH}} = \frac{\sigma^0_{VV}}{\sigma^0_{PP}}$$

(1)

where $P$ corresponds to backscatter power and $\sigma^0$ corresponds to a measure of backscatter (the *normalised radar cross-section*). For land cover with high roughness, which tends to be both a depolarising emitter and scatterer (e.g. highly topographic multiyear ice, or a melt pond under windy conditions), horizontal and vertical contributions become progressively similar as the proportion of signal from the feature causing roughness increases. On the other hand, flat (horizontal) surfaces like melt ponds tend to have $P_V > P_H$. Ratios can therefore be used as an indicator of, among other properties, the degree of scattering

cover (Woodhouse, 2006). Another property that could indicate volume scattering is the *cross-polarised ratio* as shown in Eq.2:

$$r_{cross} \frac{P_{HV}}{P_{HH}}$$

(2)

since cross-pol contribution increases with the targets responsible for volume scattering (such as once again, vegetation example). The increased moisture content of melting snow (prior to pond formation) is related to the attenuation of C-band

polarimetric backscatter such as the contribution of backscatter from the underlying ice is effectively masked. Results from the high Arctic has confirmed it for FYI advanced melt season with snow volumetric moisture content of 46%, however in marginal ice zone the onset of a freeze event is linked to strong rough surface scattering from the snow-ice interface, volume scattering from the upper ice cover and expected contributions from within the stratified and coarse-grained snow cover (Scharien et al., 2012). In fact the largest C-band co-pol and cross-pol backscatter intensities are observed for cold, snow

covered FYI during a freeze event (Scharien et al., 2012). The behaviour of C-bands VV and HH from sea ice are relatively well understood for all seasons including advanced melt, when dynamic wind-wave roughness on pond surfaces is known to cause considerable variation (Scharien et al., 2012). The reduced incidence angle dependence on the backscatter for cross



polarization channels (HV and VH) compared to the co-polarization channels (HH and VV), makes cross polarization an important asset for sea ice monitoring (Johansson et al., 2017). Combining the polarimetric information related to the scattering
mechanisms and the variation in the radiometry of the backscatter signal of the sea ice enables t further exploit the possibilities that quad-polarimetric SAR offers for sea ice identification and classification (Johansson et al., 2017).

The backscattering coefficient ($\sigma$°) is also dependent on the incidence angle, as well as the angle between the radar viewing direction and the wave orientation on the melt ponds. In fact, besides radar parameters, the melt pond's backscatter signature also depends on the electromagnetic and geometry of the surface itself (Fig.6). In the case of melt ponds, their backscattering
(while in liquid state) depends only on their surface roughness - which in turn is closely related with wind conditions (e.g. Yackel and Barber, 2000; Scharien and Yackel, 2005; Scharien et al., 2014a; Scharien et al., 2010; Fors et al., 2017). Wind conditions have a great impact, as it changes the surface roughness (e.g. Fors et al. 2017). Under calm window conditions (for instance below 3m/s), the scattering from melt ponds is mainly specular for both X and C-band frequencies, resulting in a lower backscatter intensity (Scharien et al., 2010), which results in darker pixels on a radar image (Fig.5f). The influence of
wind is itself dependent on the depth of ponds, their orientation and the surrounding topography (Scharien et al., 2010; Scharien et al., 2014b). A melt pond on sea ice, has the brightness-temperature signal of open water, meaning it will look like open water in cracks or leads between the sea ice floes.

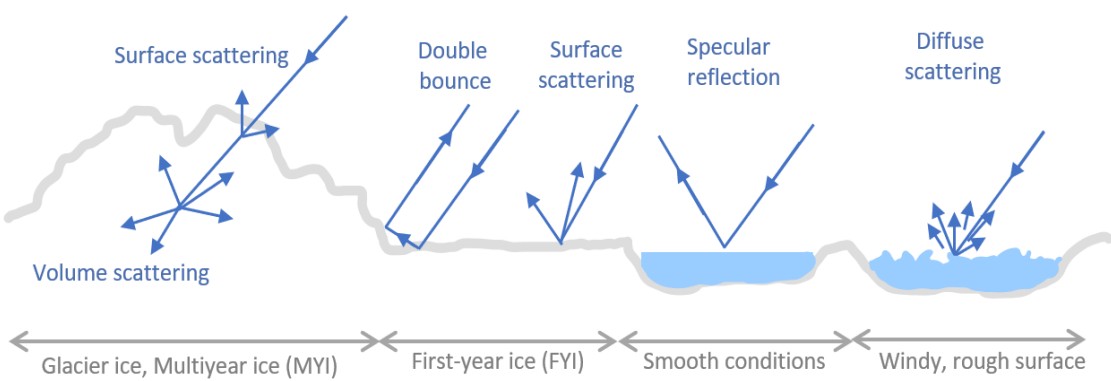

**Figure 6: Radar scattering mechanism. Different interactions of microwave radar with different sea ice types and melt pond under**
**different wind conditions**

Overall backscatter coefficient of sea ice decreases with increasing melt pond fraction, since melt ponds have a lower dielectric constant than ice, resulting in less scattering of the radar signal and a weaker backscatter coefficient, particularly at low incidence angles. As the ice surface becomes rougher due to melting and refreezing, the backscatter coefficient increases. Moreover, as melt ponds presence on sea ice affect its physical and dielectric properties, it leads to increased penetration of
the radar signal into the ice, making it difficult to accurately measure the thickness of sea ice using radar during periods of advanced melt (Scharien et al., 2010).



### 3.2 Optical and SAR technical abilities for melt pond discrimination

Optical remote sensing of snow-covered surfaces dates back to 1966, when the first map showing the extension of snow cover was created using NASA's satellite ESSA-3 (Environmental Science Services Administration), while the use of optical
monitoring of melt ponds started approximately a decade later (Grenfell and Maykut, 1977). Holt and Digby (1985) provided a thorough description of melt season processes on both FYI and MYI, using a combination of photographic, aircraft radar and satellite data. Since then, optical sensors have been used to study melt ponds for two main reasons: to monitor their shape, dynamics and spatial distribution (e.g. Tschudi et al., 2008; Wang et al., 2020; Feng et al., 2021); and for surface energy balance studies (Istomina et al., 2016; Lu et al., 2018). Optical satellite data with medium to low resolution (i.e., between 30
m to 1 km), such as the Terra/Aqua Moderate Resolution Imaging Spectroradiometer (MODIS), the ENVISAT Medium Resolution Imaging Spectrometer (MERIS) and the Landsat-7 Enhanced Thematic Mapper Plus (ETM+) multispectral images have been used to observe melt ponds in the Arctic (Markus et al., 2003; Tschudi et al., 2008; Rösel et al., 2011, Rösel et al., 2012; Istomina et al., 2015; Zege et al., 2015)(Fig.7), thanks to the distinctive spectral albedo of melt ponds. More recently, higher resolution satellites such as the Copernicus Sentinel-2 Multispectral Instrument (MSI), have been used visible and near
infrared bands for the study of melt ponds (Wang et al., 2020; Qin et al., 2021; Niehaus et al., 2022) and m-scale resolution, such as m-scale WorldView (WV) imagery (e.g. Wright and Polashenski, 2018; Tilling et al., 2020).

Notwithstanding being the most used type of sensor for melt pond studies (in comparison to SAR and passive microwave data)(Fig.7), one important constraint to optical satellites is the fact that their resolution is sometimes at the expense of swath coverage. For instance, Landsat 7 has proven to be well-suited for mapping melt ponds given its 30 m spatial resolution
(Markus et al., 2003), as melt ponds are generally less than 10 m in size. However, this satellite is not well able to perform pan-Arctic coverage in a single day as its swath width was only 185 km. On the other hand, the MODIS and MERIS, have much larger swath widths (e.g. > 1000 km), but they have a coarser resolution (250 to 1000 m). Furthermore, when using MODIS L1B products, each image has distinct striping patterns since each detector is calibrated independently (Lee et al., 2020), which inherently biases melt pond retrievals. The MSI on board Sentinel-2 has a significantly higher resolution, up to
10m for the visible bands (blue, green and red) and one of the visible and near infrared (VNIR) bands. However, its data is systematically acquired in all coastal waters up to 20 km from the shore and up to 82.8°, meaning that some areas of the Arctic Ocean are not covered by this mission as shown. Other commercial satellite constellations, such as DigitalGlobe and Pléiades, provide high resolution optical imagery but have limited polar coverage (up to 82°N) and data is only available under a licence.

Despite the use of optical data to map melt ponds such as MODIS-based melt pond retrieval (e.g. Rösel et al., 2012; Tschudi
et al., 2018), these approaches strongly depend on the representativeness of the spectral reflectance data used in spectral unmixing algorithms. Another drawback pointed out by the same authors, is the fact that since MODIS pixels used in their study had a much lower resolution, this could tend to over/underestimate melt pond coverage when a large/small number of melt ponds occupy a single MODIS pixel (Lee et al., 2020). Coarser resolution of optical data, can result in unrealistically



large melt pond fraction, which is especially problematic in the marginal ice zone and may lead to an exaggerated melt pond
fraction estimate compared to the more compact ice pack in the central Arctic. Open water and leads, which play an important
role in the overall surface albedo, can also be misidentified as melt ponds (Lee et al., 2020) due to the similarity of spectral
reflectance. And finally, although optical satellites offer the significant advantage of requiring much less energy, they are
dependent on the sun. The optical-based retrieval algorithms are vulnerable to corrections for atmospheric constituents and
influences of the viewing angles and the solar geometry. Moreover they require cloud-free conditions, limiting their
applicability in the Arctic due to the persistent cloud cover present during summer (Comiso and Kwok, 1996). In summer,
Arctic stratus clouds are often present, making it a major drawback in using optical sensors in the Arctic, as these sensors are
not able to penetrate them, decreasing the amount of acquisitions which are useful despite the high spatial coverage in higher
latitudes. To summarise, gaps in the knowledge of melt ponds' accurate coverage, size, and distribution at an Arctic-wide level
remain due to current limitations in both widespread and detailed observations of melt ponds using optical techniques.

On the other hand, satellite microwave radiometers and scatterometers can penetrate clouds, but their resolution is in general
too coarse for automated melt pond monitoring (Comiso and Kwok, 1996). SAR shows a considerable promise, but
nevertheless it requires further understanding of links between changing physical properties of the sea ice – and the observed
backscatter, specially to understand its utility during advanced melt (Yackel and Barber, 2000). Conventional SAR data (e.g.
ENVISAT ASAR and RADARSAT-1) offer a spatial resolution of the order of tens to hundreds of metres and they have shown
potential for monitoring general melt processes such as the onset of melt (Scharien et al., 2007). In addition to it, SAR also
offers the advantage to optical in the sense it does not require natural illumination and provides measurements regardless of
the weather conditions.

The synthetic aperture radars of ERS-1,2 (European Remote Sensing satellites, European Space Agency (ESA), in 1991 and
1995) were utilised to monitor the sea ice on the Northern Sea Route, ice mapping and forecasts, and to provide detailed
information on sea ice to select the ship routes. The launch of ENVISAT (Environmental Satellite, ESA) in 2002 with the
multiple polarization ability of Advanced SAR (ASAR) data represented a technological improvement of spaceborne SAR
systems allowing investigation of polarimetric scattering signatures of various sea ice types at different incidence angles.
Specifically for melt pond monitoring, spaceborne SAR has proven to be a promising approach (Scharien et al. 2014a; Han et
al., 2016; Li et al., 2017) due to their high-resolution and sensitivity to surface roughness, dielectric properties and viewing
geometry (Woodhouse, 2006; Scharien et al., 2010). As concluded by Kim et al. (2013), the size and shape of melt ponds
derived from high-resolution SAR could provide a level of detail and accuracy, compared to those obtained from aerial
photographs. However, SAR is also poised to underestimate the melt pond coverage and number density of small ponds to
some degree depending on the resolution (Huang et al. 2016). Assessment of absolute C-band backscatter magnitudes, for
satellite-scale requires work on incorporating measurements of surface roughness and its evolution during advanced melt
(Scharien et al 2010).





**Figure 7: Overview of Earth observations (EO) studies of melt ponds and MPF retrievals since 2000 with highlights to EO platform (air- or spaceborne), sensor and bands used.4 Melt pond fraction (MPF) satellite-based retrievals**



## 4 Melt pond fraction (MPF) satellite-based retrievals

Besides the importance of the knowledge of melt ponds shape, location and dynamics, one of the key aspects of interest in the context of climate studies is their actual coverage on sea ice, often referred to as melt pond fraction (MPF). MPF is an important geophysical parameter and it has been widely used in sea ice evolution and general circulation models (e.g. Flocco and Feltham, 2007; Polashenski et al., 2012). More recently it has been used to improve forecasts of seasonal ice changes and has been reported to be a good predictor for the Arctic sea ice extent in September (Flocco et al., 2012; Schröder et al., 2014; Howell et al., 2020; Ding et al., 2020a; Feng et al., 2021), which underscores the importance of monitoring the evolution of MPF over the Arctic sea ice. Melt pond fraction is defined as the ponded area relative to the sea ice, or in other words, fraction of sea ice covered by melt ponds (Webster et al., 2015; Perovich et al., 2002), and can be estimated following equation:

$$MPF = \frac{Melt\ pond}{Melt\ pond + Ice} \times 100 \tag{3}$$

Traditionally, MPF of the Arctic sea ice was measured through ship-based visual observation, and aerial photography, however, given the limited spatial coverage of ground measurements, the spatiotemporal variations in the MPF are not well captured, being satellite-based acquisitions more commonly used to derive MPF measurements. In fact, MPF estimations from optical data resulted in root mean square error (or RMSE, a common measure of the difference between values predicted and the actual observed values) of ~10%, 4-12% and 45% (Zege et al., 2015; Rösel et al., 2011 and Istomina et al., 2016, respectively) when compared to in situ observation and aerial photography.

In order to be detected, the diameter of a melt pond should be larger than the pixel size (Wang et al., 2020), meaning that the resolution of the sensor is a key aspect to consider. However, while low resolution satellite imagery cannot resolve individual melt ponds, it can provide information on MPF (e.g. Tschudi et al., 2008; Rösel et al., 2012; Rösel and Kaleschke, 2011). Optical sensors are widely used in MPF retrieval due to their capacity to measure the spectral reflectance from open water, sea ice and melt ponds (Fors et al., 2017) and also because of their strong intuitive and interpretative capability (Qin et al., 2021). The traditional methods for retrieving MPF from optical satellite imagery are unmixing the pond coverage from mixed pixels, or separating melt pond pixels from open ocean, leads and snow/bare-ice pixels, and from there generate statistics of ponded percentage of the sea ice as described in Eq.3. A summary of the different algorithms that have been used specifically for MPF retrievals (for both optical and microwave data) using traditional or more recent approaches with Artificial intelligence (AI) are later summarised in Table 3.

### 4.1 Optical traditional approaches

Regarding to the most used multispectral sensors used for MPF retrievals or melt pond classification, these have been the ETM+, on board Landsat 7 (Markus et al., 2003; Rösel and Kaleschke, 2011); the MODIS, on board Aqua and Terra (Tschudi et al., 2008; Rösel et al., 2012; Rösel and Kaleschke, 2012; Ding et al., 2020a; Lee and Stroeve, 2021); the MERIS on board





ENVISAT (Zege et al., 2015; Istomina et al., 2015); the Ocean and Land Colour Instrument (OLCI) on board of Sentinel-3 (Istomina, 2020) and more recently, the Multispectral Instrument (MSI) on board Sentinel-2 (Wang et al., 2020; Qin et al., 2021; Niehaus et al., 2022) as represented in the chronology of EO-based studies of melt ponds in Figure 7 with overall research findings summarised on Table 1. For MODIS, Rösel et al. (2012) and Tschudi et al. (2008) developed an algorithm based on

a linear spectral unmixing procedure to derive the fractional coverages from different surfaces types. The linear spectral unmixing algorithm works by decomposing the spectral signature of a mixed pixel, such as a pixel containing a combination of sea ice, snow, and melt ponds, into the contributions of individual surface types based on their known spectral reflectance. The algorithm uses a set of endmembers, which are spectra of known surface types, to estimate the fractional coverage of each surface type in the mixed pixel. In the case of melt ponds, the endmember spectrum is obtained from field measurements of

the reflectance of a melt pond surface. To estimate the fractional coverage of melt ponds, the algorithm requires knowledge of the reflectance spectra of the other surface types in the mixed pixel, such as sea ice and snow. In their studies, Rösel et al. and Tschudi et al. assumed that the reflectance of each surface type within each of the MODIS spectral bands to be constant, which allowed them to use a linear mixing model to estimate the fractional coverage of melt ponds.

Their results showed that estimates of MPF were in good agreement with ground measurements for the summer of 2004. Rösel

et al. (2012) used an AI-based approach (later described in chapter 4.2.2) to speed up the process and reduce the computation cost of the MPF estimations, resulting in one of the currently existing MPF dataset over the Arctic sea ice (from 2000 to 2011)(Table 3). Another study, by Kern et al. (2016), used a spectral unmixing approach to classify the fractions of open water, melt ponds and sea ice (using typical reflectance types as described by Rösel et al., 2012). In this study they additionally performed a linear regression between the retrieved MPF and AMSR-E retrieved sea ice concentrations to understand the

impacts of sea ice by the melt pond coverage.

However, the use of linear spectral unmixing results in large estimation uncertainty due to the prior fixed spectral reflectance, since melt pond fraction and melt pond depths change it considerably (Zege et al., 2015). It was exactly to address this problem, that Zege et al. (2015), when retrieving MPF from MERIS sensor, developed an algorithm where reflectance values were not fixed, but applied a physical model to derive them from the inherent optical properties of bare sea ice and melt ponds. A model

was developed to simulate the bidirectional reflectance factor (BRF), black-sky and white-sky over the surfaces of melt ponds, snow and ice. The BRF measures how much light is reflected by a surface at different angles and wavelengths, taking into account the direction of incoming and outgoing light as well as properties of the surface (e.g. reflectivity), which can vary significantly depending on the presence and properties of melt ponds, snow and ice. For instance, melt ponds have a higher reflectivity than bare ice, especially in the visible range, due to their smoother surface and lower absorption of solar radiation.

Istomina et al. (2015) used the same algorithm developed by Zege et al. (2015), with a main difference on the validation of the algorithm, since they conducted a more extensive validation using a larger dataset of ground-based measurements and field observations, covering a wider range of ice conditions and geographic locations. The methods of Tschudi et al. (2008) and





Rösel et al. (2012) strongly depended on the representativeness of the spectral reflectance data used in the spectral unmixing algorithms. Large errors in MPF were associated with the deviation between reference spectra and the actual melt pond reflectance. Yackel et al. (2017) retrieved melt ponds using a larger database of reference reflectance and Multiple Endmember Spectral Mixture Analysis (MESMA) to map sub-pixel fractional areas of melt ponds from MOD09 surface reflectance data. However, as is the case with spectral unmixing, the accuracy of the results strongly depends on the representativeness of reflectances for each surface type.

Wang et al. (2020) avoided the traditional fixing of reflectances approach as well, treating melt ponds as variable-reflectance, by developing a new algorithm "LinearPolar" and with increased resolution of optical data. This algorithm is based on the observation that the distribution of MPF and ice classes in the imagery can be represented by two axes: the melt pond axis and the sea ice axis. The algorithm applies a polar coordinate transformation to the distribution, using the intersection of the melt pond axis and the sea ice axis as the centre. To apply the transformation they used a combination of two spectral bands from Sentinel-2 MSI, the band 2 (blue band) and the difference between band 8 (near infrared band) and band 2, since they provide the best result based on the reported spectral albedo values of different sea ice types. The 2-D scatterplot of b2 and b28 (Fig.8a) is transformed to polar coordinates using the sea ice axis and the melt pond axis as reference axes. The angle $\theta$ between a pixel's spectral properties and the melt pond axis is used to determine the MPF, and the radial coordinate $r$ is associated with spectral variation between different sea ice types and melt pond types in the pixel (Fig.8b). To distinguish pure sea ice pixels from those with a nonzero MPF, a threshold $\theta t$ was used, and MPF is proportional to $\theta$. Mixed pixels lie between the melt pond axis and the sea ice axis and have intermediate values of MPF. The algorithm has been tested and shown to accurately estimate MPF in sea ice imagery. Their results have shown a better performance, with a 30% lower RMSE value compared to other approaches (e.g. than triangle-based algorithms by Markus et al. (2002) and principal component Principal Component Analysis (PCA)-based methods by Rösel and Kaleschke, (2011)). Qin et al. (2021) applied the same algorithm but to Landsat-8 data to determine the best band combination bands, proving that LinearPolar algorithm has also good applicability with low- or medium-resolution data, having arrived to a relative error (RE), of 53.5 and 46.6% lower than the results from Markus et al. (2002) and Rösel and Kaleschke (2011), respectively. More recently, LinearPolar has been also applied to Sentinel-2 bands 2 and 8, enabling the generation of a MPF at 10 m resolution, wily achieving a uncertainty of 6%.

Also using Sentinel-2 (using bands 2, 4 and 8) and in combination with Landsat 8, Li et al. (2020) developed a methodology to retrieve MPF by establishing a sample library by selecting numerous sample points and used the fully constrained least squares (FCLS) algorithm to retrieve the abundance of different surface features, including melt ponds. Then applying the Equation 3 earlier mentioned, they were able to calculate MPF. The accuracy of MPF retrieval by the FCLS algorithm was evaluated by comparing it with the corresponding WorldView-2 data classified data, and resulted in RMSE~0.06.





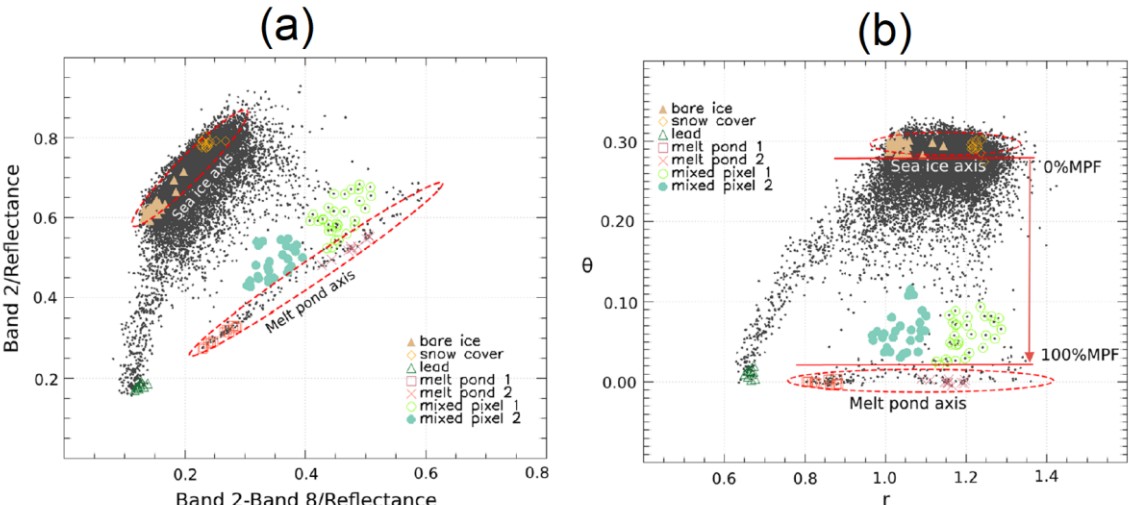

**Figure 8: "LinearPolar" 2-D scatterplots by Wang et al. (2020) of a Sentinel-2 image: (a) S2 reflectance of b2 vs b8 for selected pixels**
**(black dots refer to the coordinate value of unselected pixels); (b) S2 reflectance in the transformed new coordinate system with axes**
**θ and r (black dots refer to the coordinate value o the same unselected pixels.**

Other higher resolution studies without using satellites, such as helicopter-borne campaigns, have been conducted in the past

years (e.g. Renner et al., 2013; Divine et al., 2015; Miao et al., 2015; Webster et al., 2015; Huang et al., 2016). In the helicopter-

borne studies of melt ponds, although some have already used AI techniques, there is a prevalence for thresholding-based

approaches. For example, Divine et al. (2015) performed a three step object identification and classification procedure on

helicopter-borne imagery. First, they chose a threshold to minimise an intra-class variance, resulting in image

segmentation/binarization. Then they were able to boundary trace on the binarized images and to perform object classification

using thresholding in the red channel intensity. This revealed a homogeneous melt across the study area with a melt-pond

coverage of about 29%. The authors applied the moving block bootstrap technique to sequences of classified sea-ice images

and albedo of the four surface types to yield a regional albedo estimate. However, threshold techniques have also been used

on satellite imagery: Webster et al. (2015) developed an algorithm to classify within 4 classes (sea ice, thin ice, melt pond and

open water), through physically based thresholds which accounted for the radiometric inconsistencies in the satellite data

imagery and also taking into consideration neighbouring pixels' intensities. The outcome of this work, generated the MEDEA

MPF dataset (Table 4), which have been used in multiple studies for validation purposes for example in Ding et al., 2019; Ding

et al., 2020a; Lee et al., 2020; Peng et al., 2022 (Table 1). In another example, Huang et al. (2016) is one of the few satellite

based studies (using MODIS data), that also applied threshold. In their study, images were segmented into three classes based

on the colour difference of ice surface features: snow-covered or bared ice, melt ponds, and open leads, which was manually

selected RGB thresholds based on colour distribution histograms of each image independently.





**Table 1: Melt pond fraction remote sensing literature review from summary of main research conclusions and datasets used on melt pond classification and melt pond fraction retrievals studies on MYI and/or FYI ice types**

| AOI and/or ice types | Period | Goal & Main remarks | Remote Sensing and Campaign (Ca), Validation (V) and Comparison (Co) datasets | References |
|---|---|---|---|---|
| pan-Arctic MYI and FYI | 2000 to 2020 | To increase the temporal span of retrieval of MPF using MODIS daily data, achieving accuracies of R2 = 0.76, RMSE = 0.05. | - MODIS09GA: 1-7 bands<br>- UB-MPF and (Co): UB-OLCI, UH-MPF, BNU-MPF (Table 2)<br>- (V/Ca): ASIMPSM, MEDEA, TransArc, IceWatch and HOTRAX | Peng et al. (2022) |
| Track of MOSAI campaign | 2019 to 20210 | Retrieval of MPF using high resolution data, following the LinearPolar algorithm. MPF with a resolution of 10 m and uncertainty of 6%. MPF evolution was analysed. | - Sentinel-2 MSI: 2 and 5 bands<br>- (V): SkySat data classified with OSSP (Wright and Polashenski (2018)); (Ca): Helicopter-borne Canon EOS 1D Mark III (MOSAIC) | Niehaus et al. (2022) |
| Canadian Arctic Archipelago (CAA) | 2006 to 2018 | Retrieval of MPF based on brightness temperature. Identified 10 and 18 GHz as best channels to be used on the gradient ratio at near-shore environments. | - AMSR-E/2 brightness temperature data different channels at H-polarization | Tanaka and Scharien (2022) |
| pan-Arctic (north of 60 N) | 2017 (Mid June to late July) | Establishing a relationship between MODIS daily product and MPF. MPF estimation accuracy to an RMSE of 3.7%, compared with other models (~5%). Allowed to show MPF's seasonal cycle and compare with other MPF products and sea ice extension. | - MOD09AI (8-day): 1 - 5 bands<br>- (Co:) UH-MPF (Table 2); NSIDC (Table 4); Air temperature and sea level pressure (NOAA)) | Feng et al. (2021) |
| NW Passage locations in CAA | 2016 2018 2019 | Determined the best band combination for LinearPolar algorithm in L8 data (B2-B25). Comparison between LinearPolar and PCA | - Landsat (8 scenes): 2,3,4 and 4 bands<br>- (V): Sentinel-2 MSI (6 scenes): 2 and 8 bands | Qin et al. 2021 |
| CAA Landfast FYI | 2009 to 2018 (April) | Retrieving peak MPF with time series following and checking their predictive potential for summer sea ice. Correlations were found. | - RADARSAT-2: HH-pol, incidence angle between 20.0° and 49.3°<br>- (Ca): LiDAR (Landy et al., 2014)<br>- (Co.) UH-MPF (Rösel et al., 2012) | Howell et al. (2020) |
| pan-Arctic MYI and FYY | 7 years from to 2019 | Melt pond classification (4 classes) and MPF retrieval for TOA 20-year record data. Normalised band differences produced the best results. Classification accuracy of 85.5% and MPF's RMSE of 0.18. | - MOD02HKM: 1-4 bands<br>- (V): WorldView, NSIDC, MEDEA (Table 4)<br>- (Ca): ARKTIS-XXII-2, PS86 data | Lee et al. (2020) |





| | | | | |
|---|---|---|---|---|
| pan-Arctic MYI and FYI | 2000 to 2019 | Retrieved melt pond fraction over Arctic sea ice. RMSE below 0.1. Sea ice concentration (SIC) added as target data has a minor effect on the MPF retrieval. Created a new (longest to date) MPF dataset from 2000 to 2019. | - MOD09A1: 1-7 bands<br>- (Ca/V): HOTRAX, DLUT, TransArc, PRIC-Lei, NSIDC, NPI IceWatch and JOIS; (V) MEDEIA (Table 4)<br>- WV-MPF( Table 2) | Ding et al. (2020) |
| pan-Arctic MYI and FYI | 2000 to 2017 | Create MPF from MODIS using a E-DNN. RMSE ranged from 0.48 to 0.67 and correlation coefficient from 6 to 12% depending on MPF observations. Longest to date MPF. Tempo-spatial evolution shows increasing trends. | - MOD09A1: 1,2,3,5 bands<br>- (Ca): HOTRAX, DLUT, TransArc, PRIC-Lei, NSIDC dataset (Table 4)<br>- (V): MEDEA (Table 4) | Ding et al. (2019) |
| Canadian Arctic Archipelago (CAA), MYI and FYI | 2017 (4 days) | Improve the accuracy of MPF retrieval with a new algorithm that takes into account the variable reflectance of melt ponds. Their LinearPolar algorithm was shown to be more accurate and precise than previous methods, with a 30% lower RMSE value. | - Sentinel-2: 2 and 8 bands<br>- (Co): IceBridge DMS imagery | Wang et al. (2020) |
| CAA, MYI and FYI | 2017 (Summer) | Retrieval of MPF using fully constrained (FCLS) with optical data with high accuracy (RMSE~0.06). Evolution of melt ponds on FYI/MYI and relationship with albedo and temperature | - Sentinel-2, 2,4 and 8 bands<br>- Landsat 8 L2<br>- (V):WorldView-2 (WV2) (Table 4)<br>- ERA-Interim 2 m temperature reanalysis data | Li et al. (2020) |
| Beaufort/Chukchi Sea and central A. ocean, FYI and MYI | 2016 to 2017 (July) | Classification of pixels into four classes (undeformed ice, deformed ice, open water and ponded sea ice) and also on three colours of ponds (related to darkness). Differences between MPF and colours related to FYI vs MYI. | - (Ca): Airborne Digital Mapping System (DMS) (IceBridge NASA)<br>- (Co): AMSR2 SIC | Buckley et al. (2020) |
| FYI and MYI | 2009 2014 2016 | New algorithm Open Source Sea Ice Processing (OSSP) that classifies surface into 3 main categories: snow and bare ice, melt ponds and submerged ice, and open water, demonstrated on four sensors. Accuracy over 96%. | - WorldView (panchromatic) (75m)<br>- WorldView 8-bands (multispectral) (125m)<br>- NASA IceBridge, Canon EOS 5D DMS (25m) | Wright and Polashenski (2018) |
| Resolute Passage, Canada, Landfast FYI | 2012 | Correlations between the melt pond fractions and late-winter linear and polarimetric SAR parameters and texture measures derived from the SAR. Best RMSE of 0.09. | - Aerial photography<br>- RADARSAT-2: HH, VV, HV, VH, incidence angles from 23.1 to 42.6°<br>- (Ca): Airborne Canon G10 | Ramjan et al. (2018) |
| 10 sites in Arctic Ocean FYI | 2000 2006 2012 2018 | First attempt on MPF retrieval from hybrid-polarised compact polarization (CP) SAR imagery. Systematic overestimations due to same polarimetric characteristics as water and FYI assumed as bare ignoring effect of snow. Limitation to incidence angles larger than 35 degrees. | - RADARSAT-2: quad-pol data (VV/HH) | Li et al. (2017) |



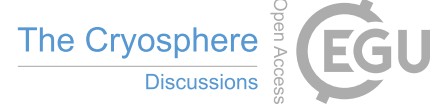

| North of Svalbard, Drifting FYI | 2012 | MPF retrieval from 4 dual-pol X-band. Best results were achieved for co-pol ratio for medium wind speeds, and VV-pol for low wind speeds. Incidence angle 29° show best results - above 40 not good. | -TerraSAR-X: HH-VV<br>- (Ca): Helicopter-borne Canon 5D Mark II and GPS/INS (ICE2012)<br>- Weather station for wind retrievals | Fors et al. (2017) |
|---|---|---|---|---|
| pan-Arctic FYI | n.a. | Estimation of MPF at the basin scale. Method provided reliable MPF for regions of high pond coverage, but overestimated MPF in regions of low pond coverage. | - MOD09<br>- (V): QuickBird data | Yackel et al. (2017) |
| NW Passage MYI and FYI | 2016 (Winter and spring) | Estimating MPF using only HH-pol $\gamma^\circ$ during with a RMSE of 0.09. Strong correlation between winter backscatter coefficient and MPF: spring MPF can be predicted based on correlation between winter backscatter and spring MPF. | - Sentinel-1 EW Mode: HH $\gamma^\circ$<br>- GeoEye-1 (GE): 4 bands | Scharien et al. (2017) |
| Beaufort Sea, CAA, Fram Strait | 2000 2001 | Retrieval of MPF from MODIS daily product, achieving RMSE of 3.91% and correlation coefficient of 0.73, outperforming previous unmixing algorithms | - MOD09GA (daily) 1-7 bands<br>- MOD09AI (8-day) 1-7 bands<br>- NSICD data (Table 4) | Liu et al. (2017) |
| CAA and Beaufort Sea | 2005 to 2014 (Jul-Aug Sep-Oct) | Estimation of MPF from comparison of AMSR-E to ship-born observations. MPF retrieved from 89 GHz provided more details in areas of high sea ice concentration. | - AMSR-E: 6.9 GHz, H-pol and 89.0 GHz, V-pol<br>- (V): MODIS MPF (Table 2)-<br>(Ca): HOTRAX (Table 4) | Tanaka et al. (2016) |
| Chukchi Sea, MYI | 2011 3 days | Classification of melt ponds, water and sea ice, although hard to discriminate melt ponds from open water. HH contributed the most with the Random Forest approach. MPF retrieval achieved RMS deviation of 2.4% | - TerraSAR-X: HH and HV-pol with 32.7° incidence angle<br>- Airborne X-band (9.3 m resolution)<br>- Aerial photographs (Kim et al. 2013) | Han et al. (2016) |
| Central Arctic FYI and MYI | 2010 (Summer) | Classification of surface into three different categories: snow-covered or bare ice, melt ponds and open leads. Further statistics on melt pond characteristics were collected. | - Helicopter-borne photograph with Canon G9 (CHINARE2010) | Huang et al. (2016) |
| pan-Arctic FYI and MYI | 2009 (June to August) | Deriving MPF from MODIS, to understand how sea ice concentration retrievals are impacted by the presence of MPF | - MOD09GA: 1,3,4 bands<br>- (Co): AMSR-E/Aqua: 16 Ghz | Kern et al. (2016) |
| Beaufort/ Chukchi Sea region FYI and MYI | 2011 | Analysed seasonal evolution of melt ponds on Arctic sea ice using, for an entire melt season on drifting first-year and multiyear sea ice. Classification into 4 classes sea ice, thin ice, melt pond and open water. | - Panchromatic satellite data (1 m)<br>- (Ca): Airborne and in situ data (NASA DISTANCE) | Webster et al. (2015) |
| pan-Arctic | 2002 to 2012 | Development of a new algorithm to retrieve MPF from optical data, without prior fixed values of spectral reflectances and accounting with bi-directional reflectance | - MERIS Level 1B: 1-15 bands<br>- (Ca):MELTEX | Zege et al. (2015) |





| | | | | |
|---|---|---|---|---|
| | | and atmospheric corrections. Errors for the case of dark ponds can exceed 50%. | | |
| FYI, MYI Pan Arctic | 2008 and 2006 | Algorithm to retrieve MPF from MERIS data. Unscreened cloud overestimates MPF before melt onset and underestimates MPF during the melt season. FYI floes results are worse due to ice drift. Ambiguities on retrieved MPFs, suggest with addition of temperature could improve results. | - MERIS Level 1B: 1-4, 8, 10, 12-14 bands<br>-(Ca/V): Barrow 2009, MELTEX 2008, NOGRAM-2 2011, NOGRAM-2 2011, C-ICE 2002, HOTRAX 2005, TransArc 2011, POL-ICE 2006 | Istomina et al. (2015) |
| Campaign CHINARE 2010 | 2010 (Summer) | Classification of high resolution data in four classes: (water, submerged ice, melt ponds and submerged ice along ice edges), shadow, and ice/snow). Overall accuracy of 95.5%. | - Aerial photographs with Canon G9 (helicopter-borne,CHINARE2010)<br>- (V/Ca): Ship-based observations | Miao et al. (2015) |
| North of Svalbard, Nansen Basin, FYI | 2012 July to August | Morphological and optical properties of a relatively thin FYI Arctic sea ice pack. 10000 classified images with homogeneous MPF. | - Helicopter-borne photograph with Canon EOS 5D ( ICE12)<br>- In situ broadband albedo measurements | Divine et al. (2015) |
| CAA Landfast FYI | 2012 May to June | Developed VV/HH-based model to retrieve MPF during 3 ponding stages. HV/HH offer more potential over VV/HH ratio, RMSE 0.05-0.07 comparable to optical approaches | - RADARSAT-2: HH, VV, HV, VH | Scharien et al. (2014) |
| North Fram Strait, Greenland, Svalbard | 2009 June-August | ENVISAT SAR-based MPF retrieval. SAR σ° mosaics were visually compared with MODIS MPF, along with spatio-temporal coinciding data. Hard to depict correlation except for smooth landfast FYI. | - ENVISAT WSM: HH-pol<br>- (Co): UH-MPF dataset, MOD09GA RBG:3-6-7 and 2-1-3 bands | Mäkynen et al. (2014) |
| Campaigns (North Svalbard and Fram Strait) | 2010 | Semi-automatic classification of melt ponds, open water, thin ice, bare ice, and submerged ice (5 classes) in combination with sea ice thickness measurements. Provided insight on relation between MPF and sea ice thickness. | - Helicopter-borne photograph with Canon EOS 350D | Renner et al. (2013) |
| Campaign Chuchi Sea | 2011 (Summer) | Mapping melt ponds using very high airborne and space-born high-resolution X-band SAR. Results were comparable with aerial photographs from previous studies. | -(Ca): Helicopter-borne X-band NanoSAR: HH-pol (KOPRI-led R/V Araon) and airborne SAR (SHEBA),<br>- (V): aerial photograph; (Co): TerraSAR-X Stripmap mode (6 m and dual-pol HH and VV) | Kim et al. (2013) |
| pan-Arctic FYI and MYI | 2008 | Estimation of MPF, and ice and water coverage and sea ice concentration for the entire Arctic region, improving Tschudi et al. (2008). Showed good agreement with observations. | - MOD09AI (8-day): 1,3,4 bands<br>- (V): MOD09AG daily: 2,3,4 bands<br>-(V/Ca): HOTRAX, NSIDC, MELTEX (Table 4) | Rösel et al. (2012) |



| | | | | |
|---|---|---|---|---|
| Northern Beaufort Sea | 2001 (July) | Comparison of MPF retrievals from Landsat and MODIS. Classification in 4 classes (open water, snow-covered ice, and two types of melt ponds). Results showed problems with saturated pixels, which is related to sun elevation and the surface type. | - Landsat 7 ETM+: 8 bands<br>- MOD09: 1,3,4 bands | Rösel and Kaleschke (2011) |
| 3 Locations Arctic Ocean | 2017 (August) | Analysis of the potential of radar backscatter data for melt pond identification using different frequencies. MPF estimates become more realistic if X- and Ku-band are used in combination with C-band. | - Helicopter-borne Multi³Scat radar: S-, C-, X-, Ku-band at HH, HV, VV and VH, from incidence angles between 20 and 60°<br>-(Co/Ca): Video data (ARKCCII/2) | Kern et al. (2010) |
| Beaufort/Chukchi Sea region | 2004 | Estimation of MPF evolution at the basin scale. Areal extension of melt ponds has increased over the study period. | - MOD09: 1-3 bands<br>- (V): UAV digital camera images | Tschudi et al. (2008) |
| 4 Arctic Ocean sites | 1999 to 2001 | Product of melt ponds statistics (2 or 3) classes image of ice, open water and melt pond and MPF of 500 m cell with 1 m resolution. Attempts to relate MPF and SIC from microwave data proved unsuccessful. | - High resolution optical satellite imagery (n.d.) | Fetterer et al. (2008) |
| Campaign Wellington Channel FYI | 1997 | Understanding the capacity of the RADARSAT-1 time series for melt pond coverage. Found correlations between the scattering coefficients and the MPF and retrieved MPF geophysical parameters. | - RADARSAT-1 (C-band): HH, several incidence angles<br>- (Co/Ca): In situ Temperature and wind velocity and aircraft video data | Yackel and Barber (2000) |

## 4.2 Radar data traditional approaches

Although MPF optical-based retrieval outnumbers radar-based MPF retrievals, few MPF retrievals have been developed as
well for radar sensors, such as SAR (e.g. Fors et al., 2017; Li et al., 2017; Scharien et al., 2017; Howell et al., 2020) followed
by passive microwave data (e.g. Tanaka et al., 2016; Tanaka and Scharien, 2022) (Table 1). The first study using radar for
MPF retrieval dates back to 1997, when SAR aboard ERS-1 was used by Jeffries et al. (1997) to retrieve MPF over MYI,
however without considering wind which limited the validity of the model. Wind as a key parameter was found later, as Yackel
and Barber (2000) demonstrated a significant relation between HH intensity and MPF for strong and intermediate winds using
SAR aboard RADARSAT-1. In this study, the authors used a geophysical inversion, inverting the SAR signal to obtain
estimates of physical properties of melt ponds (and sea ice), such as MPF, surface roughness or dielectric constant. Ever since,
radar has been gaining a lot of attention, given its clear advantage of being all-weather and day plus night operation, offering
independence of cloud cover, illumination and atmospheric constituents. In what regards the polarization of sensor, there as



been several studies mainly focusing either on single-polarization SAR, transmitting and receiving either vertical (VV) or
horizontal (HH) polarised waves; (e.g. Tanaka et al., 2016; Scharien et al., 2017); dual-polarization (e.g. Kim et al. 2013; Han
et al., 2016), while attempts of cross polarization and polarization ratios have been showing promising results (e.g. Kern et al.,
2010; Scharien et al. 2014; Fors et al., 2017). Yackel and Barber (2000) investigated the evolution of melt pond coverage in
Canadian islands using RADARSAT-1 SAR images, and determined that backscattering coefficient was significantly
correlated with MPF. Howell et al. (2020) used RADARSAT-2 to map the MPF on Canadian islands from 2009 to 2018 based
on this relationship. In fact, this was one of the few studies to have used radar based for direct MPF retrieval (Mäkynen et al.,
2014; Scharien et al., 2017; Howell et al., 2020).

Scharien et al. (2014) developed a model to retrieve MPF based on RADARSAT-2. model during three distinct ponding stages,
through the following Equation:


$$MPF = \frac{(VV/HH)}{A[e^{B\theta}]} \tag{5}$$

Where $\Theta$ is within the range of 25 and 55° and VV/HH is in dB. A and B are model coefficients (0.3869 and 0.0571,
respectively) which were retrieved from fitting to scatterometer observation of ponds using a non-linear least-square method.
They authors achieved RMSEs values (when comparing to aerial data) of 0.05 to 0.07. Also for C-band but with higher
resolution, Scharien et al. (2017) estimated MPF from Sentinel-1 imagery by backscatter with GeoEye-1 imagery. Instead of
using σ° (i.e. sigma naught - which is most used, the ratio of the backscatter power to the power of the incidence radar signal,
and a measure of how much radar energy is reflected back to the radar from the target), they used $\gamma°$ (i.e. gamma naught),
which is the radar reflectivity factor, which is a measure of the power of the backscattered radar signal per unit volume of the
target, which is also expressed in decibels and a measure of the radar reflectivity of the target, meaning it is also a measure of
backscattered power from a radar signal, but normalised differently. Their approach involved relating the winter period HH
$\gamma°$ to peak MPF observation in 1.7 m spatial resolution GeoEye-1 imagery, from spatially coincident image segments that
represented homogeneous FYI and MYI regions. The result was that $\gamma°$ can be converted to MPF using the following Equation
4:


$$MPF = -0.221 - 0.041(\gamma°) \tag{4}$$

Scharien et al. (2017) also reported a strong correlation between the Sentinel-1 winter ice backscattering coefficient and spring
MPF with a correlation of 0.85.. Howell et al. (2020) following a modified approach, retrieved a RADARSAT-2-based MPF
for the Canadian Arctic Archipelago (CCA). To overcome the challenge from incidence angle variability, they averaged it by





taking advantage of the overlapping imagery within the region, and applying the conversion by Scharien et al. (2017), they produced a RADARSAT-2 MPF, calculating the mean MPF for each overlapping pixel, reaching RMSE values of 0.10-0.12. Also Mäkynen et al. (2014) compared the backscattering coefficient of ENVISAT wide swath mode (WSM) SAR images in

HH polarization to the melt pond fraction of the Arctic sea ice derived from MODIS-based MPF products by Rösel et al., 2012 (UH-MPF dataset in Table 3). The researchers used three methodologies to analyse the relationship: visual comparison of MODIS-MPF charts, MODIS RGB images, and SAR mosaic, extraction of σ∘ and MPF time series, and the study of statistical relationships between σ∘ and MPF. They found that there was little correspondence between the increase and decrease of MPF and the σ∘ statistics, indicating that MPF estimation from ENVISAT images is not possible, probably due to the low resolution

of the WSM Images (100 m). Also using RADARSAT-2, and based on retrieval methods from Scharien et al. (2014b), Li et al. (2017), made the first attempt on retrieving the melt pond fraction from hybrid-polarised compact polarization (CP) SAR imagery (VV/HH), by establishing an empirical relationship between the co-pol ratio and CP parameters, using a tilted-Bragg scattering model and the RS-2 qual-pod SAR observations, reaching an overall good agreement however with systematic overestimations.


A small number of studies have been conducted to understand the potential of MPF retrieval using higher frequencies like polarimetric X-band SAR (8-12 GHz) (Kern et al., 2010; Kim et al., 2013; Han et al., 2016; Fors et al., 2017). TerraSAR-X has been explored by Han et al. (2016) and Kim et al. (2013) for melt ponds studies over multiyear ice and Fors et al. (2017) on first-year ice. Kim et al. (2013) used X-band dual polarization HH and VV, from helicopter borne SAR sensor, to delineate

ice and melt ponds. Using a processing software based on a combination of multiscale segmentation and aggregations methods, the ice and melt ponds were then exported as binarized images of water and ice, upon which MPF statistics were possible to be estimated. Visual comparison with aerial photographs showed a good agreement, however they also concluded that only large melt ponds were detectable (leading to the underestimation of MPF).

A common approach for MPF retrieval of radar is also correlations analysis between X-band backscatter values and MPF based on optical data (from airborne or satellite-based studies (Fors et al., 2017; Ramjan et al., 2018). Fors et al. (2017) co-located helicopter data which delivered MPF information (following Divine et al. (2015) with high resolution TerraSAR-X scene with a HH-VV combination, calculating the statistical dependence between the extracted SAR features and the corresponding MPF microwave polarization signature values, using the non-parametric Spearman's correlation coefficient.

This statistical measure allows for the quantification of the strength and direction of relationship between two variables which allow handling non-linear relationships. Although this is not a form of regression, regression can be then used to model the relationship between two variables, given the correlation coefficient. Hence, the authors fitted two regression from the correlation results. The estimated MPF distribution was compared and evaluated towards the observed MPF from the helicopter flights. This study highlighted the diversity of results which comes from different incidence angles (described in Table 1) or

wind speed conditions, which later highlighted in chapter 4.3. Ramjan et al. (2018) generated MPF data from aerial imagery,





which classified using AI methods (following Scharien et al., 2014b). They employed multivariate linear regression to generate combined backscatter, polarimetric parameter, and texture-based models to predict MPF. Spearman's rank correlation analysis was also used to assess correlation between parameters and with MPF while accounting for any non-normal distribution, the best results from nine linear, polarimetric and 72 parameters at different incidence angles arrive at RMSE of 0.09.


While Fors et al. (2017) concluded that MPF influences the signature of several X-band features, being the strongest correlations between R(VV/HH) and VV intensity. Han et al. (2016), whose AI-based method is later described in chapter 4.3.1, concluded that spatial texture HH contributed the most and that average and standard deviation of polarimetric features improved results. On the other hand, Kern et al. (2010), used several frequencies for MPF retrieval on MYI in the Arctic ocean,

showing promising results when combining C-bad frequency (5.4 GHz), with Ku (17.2 GHz) and X (9.6 GHz). Results suggested that MPF estimates become more realistic if frequencies like X and Ku-band are used in combination with C-band. A comparison between MPF retrievals from optical and radar has been performed by Mäkynen et al. (2014), who compared MODIS and ENVISAT ASAR satellite scenes, for both FYI and MYI, concluding that MPF estimation was not possible with the dataset.


Regarding passive microwave data, Tanaka et al. (2016), used AMSR-E/2 brightness temperature data to estimate MPF through linear regression between gradient ratios from H-polarization and V-polarization and ship-based observations, by analysing with linear regression the brightness histograms of the images obtained from a forward-looking camera in ship observations. The analysis of the peaks in the curves fit to histograms allowed the identification of up to three surface

conditions, i.e. open water, sea ice and melt ponds. More recently, the same approach, using other channels (Tanaka and Scharien, 2022) delivered better results with an increased resolution of the MPF product.

**4.3 AI-based techniques for MPF retrievals and melt ponds monitoring**

Artificial Intelligence (AI) has seen a considerable increase within the EO community for the generation and enhancement of

digital images captured from planes or satellites (Khelifi and Mignotte, 2020; Sun et al., 2022). The earlier studies of melt ponds employing AI techniques (e.g. Kern et al., 2010; Miao et al., 2015) used Machine Learning (ML). ML is a subfield of AI that uses algorithms and data to help a machine learn without instruction. Some of its first applications on melt pond studies were to perform classification tasks to generate surface type maps (Fetterer et al., 2008; Miao et al., 2015)(Table 1), where each pixel is segregated into different classes (e.g. ice, snow, melt pond or open water). From classified maps, some authors

calculated MPF statistics as expressed in Equation 3 (e.g. Fetterer et al., 2008; Rösel and Kaleschke, 2011; Han et al., 2016). Table 3, offers a summary of the main techniques used for melt pond classifications or melt pond fraction retrievals. It discriminates both goals and techniques are discriminated segregated by type of data (optical and microwave). Additionally,





ML and Deep Learning algorithms, identified with ANN (from Artificial Neural Network) are also highlighted and discriminated from one another.


**Table 3: Summary of main traditional algorithms and AI (machine learning and ANN) approaches used for EO-based studies on melt ponds classification, parameterization and MPF retrievals**

| | Algorithms/Techniques | Main goal | References |
|---|---|---|---|
| Optical data | Spectral unmixing (fixed spectral) | MPF retrieval + evolution | Tschudi et al. (2008); Rösel et al. (2012) Kern et al. (2016) |
| | ANN: Genetic algorithm optimised back-propagation | MPF retrieval | Peng et al. (2022) |
| | ANN: Auto encoder | MPF retrievals | Feng et al. (2021) |
| | ANN: Multi-Neural Network (MNN) + Multinomial logistic regression (MLR) | MP classification + MPF retrieval | Lee et al. (2020); Lee and Stroeve (2021) |
| | ANN: Ensemble-Based DL network (DNN) | MPF retrievals | Ding et al. (2019); Ding et al. (2020a) |
| | ANN: Multi-layer neural network architecture | MPF retrievals + evolution | Rösel et al. (2012); Liu et al. (2017) |
| | ML: Image segmentation into cluster + Random Forest classification of clusters | MP classification | Miao et al. (2015) |
| | ML: Image segmentation into binary + Classification based on thresholding | MP classification | Divine et al. (2015) |
| | LinearPolar algorithm | MP classification + MPF retrieval | Wang et al. (2020); Niehaus et al. (2022); Qin et al. (2021) |
| | Fully constrained least-squares (FCLS) algorithm | MPF retrievals | Li et al. (2020) |
| | Thresholding pixel values | MP classification + MPF retrieval | Webster et al. (2015); Huang et al. (2016); Buckley et al. (2020) |
| | ML: Principal Component analysis (PCA) | MP classification + MPF retrieval | Rösel and Kaleschke (2011) |
| | ML: Classification Supervised maximum likelihood + stats | MP classification + MPF retrieval | Fetterer et al. (2008) |
| | ML: Semi-automatic classification | MP classification | Renner et al. (2013) |
| | Melt Pond Detector algorithm (MPD) | MPF retrieval | Zege et al. (2015); Istomina et al. (2015) |
| Microwave data | Conversion backscatter to MPF | MPF retrieval + evolution | Scharien et al. (2014b); Scharien et al. (2017); Howell et al. (2020) |
| | Correlation between backscatter parameters and MPF products | MPF retrieval | Fors et al. (2017); Ramjan et al. (2018) |
| | ML: decision trees and random forest | MP classification | Han et al. (2016) |





| ML: Classification (Bayesian max likelihood) of k-mean clustered values | MP classification | Kern et al. (2010) |
|---|---|---|
| Comparison between SAR σ° or and MODIS-based MPF | MP classification + MPF retrieval | Mäkynen et al. (2014); |
| Linear regression: AMSR-E/2 and MPF | MPF retrieval | Tanaka et al. (2016); Tanaka and Scharien (2022) |
| Geophysical inversion extract MPF | MP parameters retrieved | Yackel and Barber (2000) |
| Delineation with software | MP classification + MPF retrievals | Kim et al. (2013) |

### 4.3.1 Machine learning-based studies

There has been a growing number of studies applying machine learning-based approaches to MPF retrievals (e.g. Wright, 2020; Liu et al., 2017; Ding et al., 2020a; Feng et al., 2021; Peng et al., 2022). Wright (2020) proposed that the accuracy of the machine learning methods is better than that of the linear spectral unmixing due to the error caused by fixed reflectance feature (e.g. method developed by Rösel et al., 2012). These findings are corroborated by Liu et al. (2017) and Wright and Polashenski (2020), who concluded that machine learning could improve melt pond retrievals from MODIS over current

spectral unmixing techniques. After the most traditional techniques for melt ponds studies, such as thresholding and spectral unmixing, ML started to be used for melt pond classifications, for both optical (e.g. Fetterer et al., 2008; Rösel and Kaleschke, 2011) and microwave data (Kern et al., 2010; Han et al., 2016).

Fetterer et al. (2008) used supervised maximum likelihood classification (one of the most popular methods of remote sensing classification). In summary, a maximum likelihood classification algorithm uses a training dataset, of labelled data (in this

case the type of class to which each pixel belongs, e.g. if melt pond, or bare ice) and estimates the probability distribution of the data for each class. This is done by calculating the mean and covariance of the data for each class, which represents the statistical properties of the data. Once the statistical properties of each class have been estimated, the algorithm can use this information to classify new, unlabelled data (Bishop, 2006). To classify the new data, the algorithm calculates the likelihood of the data belonging to each class based on the estimated statistical properties. The class with the highest likelihood is then

assigned to the new data point. Fetterer applied this algorithm to create surface type maps based on high-resolution visible images from the National Snow and Ice Data Center (NSIDC) and from there calculated pond coverage statistics retrieving then MPF results. Kern et al. (2010) used Bayesian maximum likelihood (BML), a similar approach but which incorporates prior knowledge about the distribution of the classes. It uses Bayes Theorem to estimate the posterior probability of a class given the data, taking into account both the likelihood and the prior probabilities of the class (Bishop, 2006). The BML

classifier was then used to perform a classification within 4 types (old ice, nilas, open water and melt ponds). In regards to helicopter-borne acquisitions, Renner et al. (2013), developed a semi-automatic classification algorithm (based on spectral and textural features in normalised images) with five ice classes: open water, thin ice, bare ice, melt ponds and submerged ice.



Han et al. (2016) used popular machine learning algorithms, namely decision trees (DT) and random forest (RF) for melt pond detection, using TerraSAR-X dual polarization SAR with mid-incidence angle. A DT is a machine learning algorithm that creates a tree-like model of decisions and their possible consequences. It works by recursively splitting the data based on the feature that provides the most information gain until a stopping criterion is met. At each split, the algorithm chooses the feature and threshold that best separates the data based on the specific criterion Quinlan (1986), whereas a RF is an algorithm that combines multiple decision trees to make a prediction by training each decision tree on a random subset of the available data and features (Breiman, 2001), and then aggregating their predictions to produce a final result. Polarimetric parameters alone (using DT and RF) were not effective to discriminate from open water due to similar polarimetric signatures. HH backscattering coefficient was identified as the variable contributing the most, and its spatial standard deviation was the next most contributing one to the classification of open water, sea ice and melt ponds using RF model.

Miao et al. (2015) developed an object-based classification algorithm to automatically extract sea ice and melt ponds from aerial photographs. The algorithm achieved an overall classification accuracy of 95.5% for four general classes (water, submerged ice – which included melt ponds, shadow and ice/snow) and a producer and user's accuracy of 90.8 and 91.8%, respectively. On this aerial photography-based classification algorithm, a first step of image segmentation grouping the neighbouring  pixels into objects according to the similarity of spectral and textural information, followed by a random forest (RF) ensemble classifier, distinguishing the four general classes. On a later step the polygon neighbour analyses further separating melt ponds and submerged ice. Following the same approach of Miao et al. (2015), Wright and Polashenski (2018) performed image segmentation dividing the image into objects which were then classified into 3 main categories: snow and bare ice, melt ponds and submerge ice and open water, and achieved a slight higher accuracy of 96%.

While supervised learning methods, a subcategory of AI defined by its use of labelled dataset to train algorithm to classify or predict outcomes, have been the most extensively used approach for classification of melt ponds or prediction of MPF, unsupervised techniques, an AI algorithm defined by the ability of identifying patterns in datasets contain data points data are neither classified or labelled, have also been used. For instance, Rösel and Kaleschke (2011) used Principal Component Analysis (PCA), for melt pond classifications. PCA is a widely used unsupervised learning technique for reducing the dimensionality of high-dimensional data by finding a lower-dimensional representation that preserves as much of the variability in the original data as possible. In PCA, the data is transformed into a new coordinate system where the new axes are chosen to align with the directions of maximum variance in the original data (Jolliffe, 1985). This statistical technique was used to reduce the complexity of the dataset by transforming the original variables into a smaller set of uncorrelated variables, which separated different pixels, i.e. water and ice from melt ponds, retrieving melt ponds fractions, from Landsat and MODIS data. In some cases, the use of unsupervised learning techniques ,comes as a first step on the research workflow. For instance Kern et al. (2010) applied a k-means clustering algorithm to determine the number and type of classes to predict with a classification algorithm. This algorithm aims to partition a given dataset into $k$ clusters, where $k$ is a predetermined number, in this case, the number of classes. The algorithm works by iteratively assigning each data point, in this case, pixel values, to the





nearest cluster centroid and then updating the centroid based on the newly assigned data point (Bishop, 2006). K-mean allowed to provide the initial cluster centres using averaged normalised radar in multiple frequency bands and cross polarizations retrieved by the helicopter-borne radar data. This cluster centres were used on a classification based on Bayesian maximum likelihood (BML) classifier based on the Wishart distribution of the covariance matrix was carried out using four surface types:
old ice, nilas, open water, and melt ponds.

### 4.3.2 Deep learning-based studies

Another branch of AI, Deep Learning (DL) is also getting a growing interest in the EO-based studies of melt ponds, especially in the past years (Table 3). DL is itself a subset of ML that is based on artificial neural networks (ANNs), which, through
learning from large amounts of data, can lead to better data representation learning. An ANN is composed of interconnected nodes or artificial neurons organised into layers (Fig.9), where the number of neurons and layers can be adjusted to optimise performances, allowing for learning representations of data samples with several ranges of abstraction levels (Khelifi and Mignotte, 2020). The recent use of ANNs to retrieve sea ice parameters (e.g. Rösel et al., 2012; Liu et al., 2017) has been showing the potential of ANN to learn the complex relationship between sea ice parameters and input data. ANNs have been
used to retrieve directly MPF values from optical data (e.g. Liu et al., 2017; Ding et al., 2019; Ding et al., 2020a; Feng et al., 2021; Peng et al., 2022) and proving to hold a great promise to address the challenge of retrieving MPF.

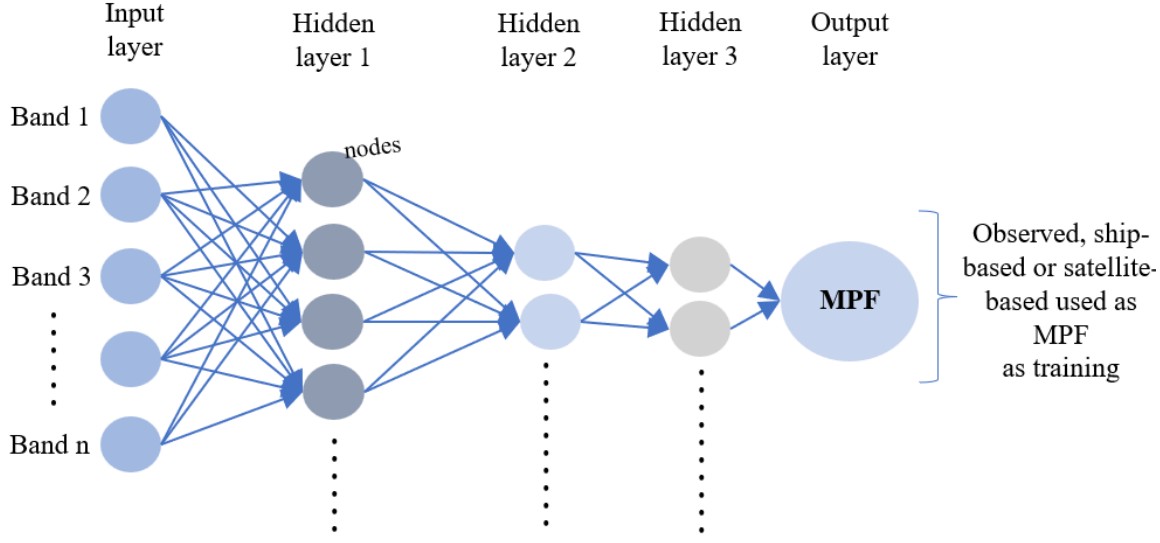

**Figure 9: Example of a multi-layer perceptron (MLP), or multi-layer network**





Rösel et al. (2012) used MODIS weekly daily reflectance (MOD09AI) to retrieve MPF, using one of the most basic forms of deep learning, a multi-layer neural network also called multi-layer perceptron (MLP), to speed up spectral unmixing of MODIS data. Their ANN, with 2 hidden layers, 9 and 27 neurons with 5000 learning steps, used as input data MODIS and output (i.e. the image that they want to generate or predict) - spectral unmixing MODIS of three classes. However in this approach since a composite of 8 days is composed of selected pixels from the daily acquisitions, especially with low cloud cover fraction and

other atmospheric influences, the generated product may represent the condition of a specific day in one week. More recent studies, using deep learning, have been applied to MODIS data for MPF retrievals which are worthy to cross-compare given their similar nature however with different approaches or results (Liu et al., 2017; Ding et al., 2019; Ding et al., 2020a; Lee et al., 2020; Feng et al., 2021; Peng et al., 2022). Liu et al. (2017) used as input data daily MODIS data product (MOD09GA) from band 1 to band 7, which wavelengths are 620-670 nm, 841-876 nm, 459-479 nm, 545-565 nm, 1230-1250 nm, 1628-

1652 nm, and 2105-2155 nm, respectively. The model, a multi-layer neural network  generated a MPF achieving a RMSE of 3.91% and a correlation coefficient of 0.73, which outperforms previous unmixing algorithms. Liu et al. also identified the importance of each band, being in decreasing order 2, 3, 1,4, 6, 5 and 7, which supports the traditional choice of selecting bands 1-3 for unmixing algorithms.

While the previous studies used atmospherically corrected MODIS reflectance data (i.e. MOD09), Lee et al., (2020), selected

to use instead of using MOD02HKM, which is Top-Of-Atmosphere (TOA) since as justified by the authors, MOD09 are not optimised for the polar regions. These authors also used a two-step approach for melt pond classification and MPF retrievals. Lee et al. (2020) used a Multi-Neural Network (MNN) to generated MODIS data within four classes: snow/thick ice, dark/thin ice, melt ponds/submerged ice, and ocean and a Multinomial Logistic Regression (MLR) for MPF predictions. The MNN model, with 3 hidden layers and 10 neurons, takes six input layers of MOD02HKM as input data and takes as output

WorldView classified data (with random forest following Wright and Polashenski, 2019). After classifying WorldView data using the random forest algorithm from Wright and Polashenski (2018), the MLR was used to predict MPF, based on ship and satellite-based reference MPF. For the MLR, MODIS-derived melt pond data were averaged for two days before and after the target date for validation. The MNN-derived melt pond classification reached 85% and the MLR-derived melt pond fraction predictions reached an RMSE of 0.18.

The most recent studies using deep learning for MODIS-derived melt pond fractions used more complex DL algorithms. Ding et al. (2019) and Ding et al. (2020a), used more complex ANN, their model an ensemble-based deep learning network (E-DNN) is more complex than the MLP used by Liu et al. and Rösel et al. The E-DNN combines multiple individual neural networks by combining the predictions of several individual models which can lead to improved performances.

In both studies, the E-DNN had 3 hidden layers, with 25, 35 and 45 neurons, respectively. Unlike Liu et al. (2017) study, Ding

et al. (2019) and Ding et al. (2020a) use weekly MODIS products (MOD09A1) as input data, however, Ding et al. (2019) used only bands 1,2,3 and 5 for input data. When compared to results from Rösel et al. (2012), Ding et al. (2019) shows a relatively





higher correlation of 0.62 and smaller RMSE 7.9% for the MEDEA MPF (generated by Webster et al., 2015, see Table 4). Ding et al. (2020a) on the other hand, used all 7 bands, corroborating as Liu et al.(2017) findings that regardless of the importance of some bands over others, results improve with the use of all the bands. In the model developed by Ding et al.

(2020a) results have also improved with average RMSE below 0.1 against most of the MPF observations datasets used. This approach also included sea ice concentration and they concluded that it had little impact. The improved results from these two studies suggest that the use of an E-DNN is yielding better results than a MLP to this end. Despite the achievements, it should be noted that in both studies, MODIS 8-days reflectivity data is used as input data, and trains the model to output daily MPF, which would introduce new errors. Therefore, for the machine learning model, appropriate features and labels need to be

matched and the performance of the machine learning model needs to be evaluated.

Feng et al. (2021) used a stacked autoencoder (SAE). A SAE is a type of ANN that is designed to learn a compressed representation of input data through a series of encoding and decoding steps. The network consists of multiple layers of autoencoder neural networks, where each layer takes the output from the previous layer as input, gradually reducing the dimensionality of the data, using bands 1 to 5 of MOD09GA the authors achieved improved accuracies (RMSE of 3.7%) in

respect to previous models (e.g. the first MPF-generated with DL by Rösel et al. 2012). Also based on MOD09GA dataset, Peng et al. (2022), used a genetic algorithm optimised back-propagation neural network (GA-BPNN) model and filled gaps using a statistical-based temporal filter, achieving a RMSE of 0.05. The study conducted extensive experiments to determine the influence of the total number of neurons and the neural network's structure on the prediction accuracy, which determined the optimal hidden layers, and neurons to occur the minimum values of RMSE.

The differences and advantages from the potential of using ANN, over machine learning approaches (e.g. RF or DT) for other traditional methods (e.g. spectral unmixing or thresholding), lie in the fact that some deep learning approaches allow the generation of MPF maps directly (e.g. Liu et al., 2017), instead of creating a classified or binary map, from which later it is computed statistics in order to obtain the MPF (e.g. Buckley et al., 2020; Niehaus et al., 2022) as illustrated in Figure 10. Another key factor is that in machine learning, the algorithm needs to be trained how to make an accurate prediction, through

feature extraction, which in this case would be data provision with identification of pixels which belong to the class of melt ponds or ice, in other words, 'labelled data'. In deep learning however, the algorithm can learn how to make an accurate prediction through its own data processing thanks to the ANN structure (Fig.9). Finally, deep learning models allow us to learn the spatiotemporal relationship between variables, which given the complexity of the nature of melt pond formation and evolution holds a great promise for future research.



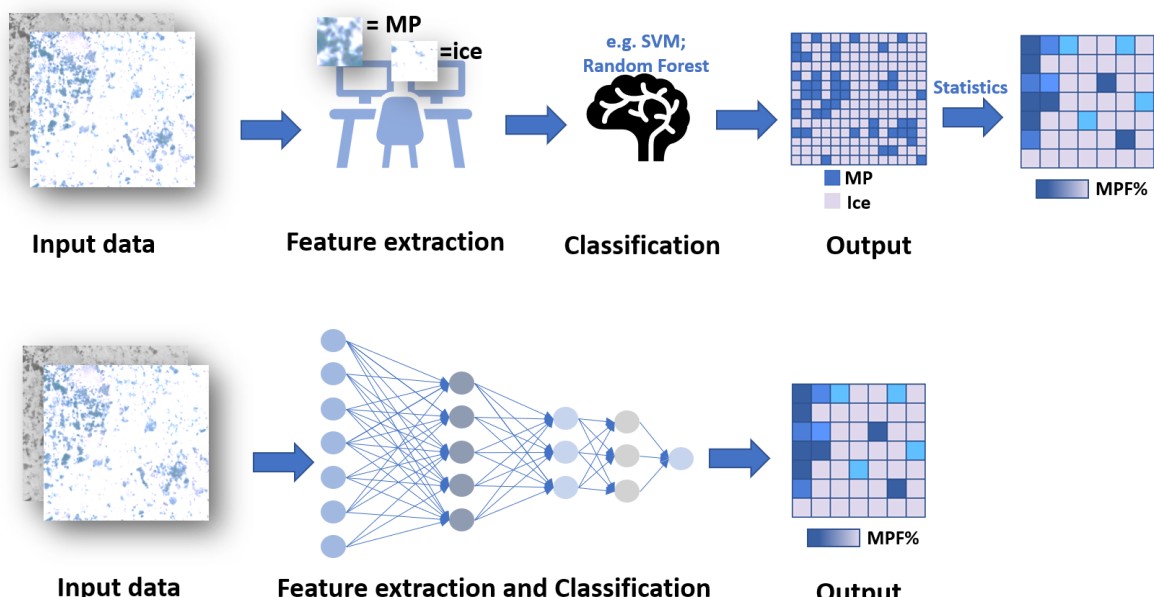


**Figure 10: Comparison of machine learning vs deep learning approaches for the generation of Melt Pond Fraction Data**

Regarding the study of evolution of melt ponds, the studies by Rösel et al. (2012) and Tschudi et al. (2008), concluded that MPF is strongly correlated with the decline of sea ice extent and thickness. Melt pond fraction increases with decreasing ice
thickness, and the spatial distribution of melt ponds is closely related to the topography of the ice surface. Webster et al. (2015)analysed seasonal evolution of melt ponds on Arctic sea ice using 1 m resolution panchromatic satellite imagery paired with airborne and in situ data for an entire melt season on drifting first-year and multiyear sea ice, developing an algorithm to classify the imagery into sea ice, thin ice, melt pond and open water classes on two contrasting ice types: first-year and multiyear ice. Sankelo et al. (2010), used an iterative image classification method to analyse the formation and temporal
evolution of melt ponds, using digital photographs to distinguish from melt ponds and other surface types (such as ice or snow). More recently work has been done to monitor the evolution of melt ponds on FYI and MYI in the Canadian Arctic with optical satellite data using optical data, namely, Sentinel-2 and Landsat 8 reflectance data (Li et al., 2020), which also looked at their evolution relationship with other variables, namely, air temperature and albedo. Ding et al. (2019) concluded, based on a time span of a 18-year studying period (2000 to 2017), that MPF evolution also presents some differences between MYI and FYI.
The growth rate in FYI is greater than on MYI which could be partly due to the fact that albedo on MYI decreases slower than on FYI in the early stages of ice melting. Ding et al.'s findings were consistent with previous research, however their study shows that the decay of MPF occurs in late July, which is about half a month later than that of previous research. Their study has also shown a large year-to-year variability of MPF, and the interannual variability after 2007 appears to be larger than that



before 2007. The year 2012 has the largest positive MPF anomaly (4.0%), and there is an increasing trend of the MPF from
2000 to 2017, but it is not statistically significant.

Rösel and Kaleschke (2012) analysed the temporal and spatial distribution of melt pond fraction in the Arctic throughout 12
years (2000-2011) using MODIS data (following Rösel et al.'s (2012) approach) and concluded that besides a negative trend
related to the declining sea ice extent, during years of extreme loss there was an increase of maximum of the coverage of melt
ponds. The decline in MPF is also evident when analysing the decreasing trend on the MODIS-Peng MPF dataset (Peng, et
al., 2022). The significant decrease can be explained by the decline in the sea ice extent. Ding et al. (2019) analysed the MPF
generated over a longer period, between 2000 and 2017 and concluded that MPF exhibited an asymmetrical growth and decay.
MPF showed growth during the first stage of its evolution (ranging approximately 8 to 12% throughout May), increasing
substantially in June (reaching approximately 23% by the end of the month), after which the growth slowed down reaching a
peak in late July. Since then MPF decreased until early September (reaching ~14% ). The findings from Ding et al. (2019) also
noted that the interannual variability of the MPF during the decay period is much larger than that during the growth period.
This could be linked to the alternating thaw and refreeze due to changes of surface temperature, rainfall and snowfall (Rösel
and Kaleschke, 2011; Rösel et al., 2012; Istomina et al., 2015).


### 4.4 Satellite-based MPF public datasets

Despite several melt pond-related datasets, currently there are very few satellite-based MPF datasets available that cover the
whole arctic and for more than one year (Table 2). Each MPF is identified with the sensor name, followed by the first author,
and within parentheses another name that other authors have referred to them. With exception of one, all are based on optical
medium resolution (the great majority, on MODIS data). One of the first MPF datasets, hereafter referred to as MODIS-Rösel
MPF (UH-MPF), was developed by researchers at the Hamburg University spanning from 2000 to 2011 (Rösel et al., 2012;
Rösel et al., 2015). For the similar time span, and provided by the University of Bremen, there is the hereafter called MERIS-
Istomina MPF (named as MERIS MPF V1.5 by the authors (Istomina et al., 2020) and refer to as UB-MPF in other studies).
By the same authors, a new OLCI-based MPF dataset hereafter referred as OLCI-Istomina MPF (named as OLCI MPF 1.5 by
the authors and UB-OLCI in other studies) was released, covering all months from May to September (and first week of
October for the year 2021). This dataset uses the same retrieval algorithm as the one used to generate MERIS-Istomina MPF,
however it is a recent version with a much wider and recent time span, covering from 2017 to the present. Ding et al. (2020a)
developed another MODIS-based MPF (or BNU-MPF), hereafter referred to as MODIS-Ding MPF dataset, covering a longer
period - from 2000 to 2019, from May to September,. The MODIS-Lee MPF dataset, covers a similar period (one year longer)
from June of the year 2000 until the end of August 2020, with a slighter wider coverage (regarding latitude) (Lee and Stroeve,
2021) and a higher resolution comparatively to all MPF datasets mentioned so far. The two most recent datasets are MODIS-



Peng MPF (or NENU-MPF) by Peng et al., (2022) and MSI-Niehaus MPF dataset by Niehaus et al. (2022). While the first covers from May to September, from 2000 to 2020. The MSI-based MPF dataset covers only the months from June to August, and given Sentinel-2 coverage, its spatial coverage goes only up to 82.3°.


**Table 2: Open pan-Arctic multi-year melt pond fraction (MPF) datasets**

| Dataset Name | MODIS-Rösel MPF (*UH-MPF*) | MERIS-Istomina MPF (*UB-MPF*) | OLCI-Istomina MPF (*UB-OLCI*) | MODIS-Lee MPF | MODIS-Ding MPF (*BNU-MPF*) | MODIS-Peng MPF (*NENU-MPF*) |
|---|---|---|---|---|---|---|
| Example preview | | | | | | |
| Time span | 2000 - 2011 | 2002-2011 | 2017 - now | 2000 - 2020 | 2000 - 2019 | 2000-2020 |
| Coverage | Lat .> 60° | Lat .> 60° | Lat .> 60° | Lat. > 57.8 | Lat .> 60° | Lat .> 60° |
| Spatial res. | 12.5 km | 12.5km | 12.5km | 1 km < 10 km | 12.5 km | 12.5 km |
| Temp. res | Weekly | Daily | Daily | Daily, Weekly, Monthly | Weekly | Daily |
| Data access | Hamburg University Portal https://www.cen.uni-hamburg.de/en/icdc/data/cryosphere/arctic-meltponds.html#beschreibung | University Bremen Portal https://seaice.uni-bremen.de/melt-ponds/ | University Bremen Portal https://seaice.uni-bremen.de/melt-ponds/ | British Antarctic Survey Portal https://ramadda.data.bas.ac.uk/repository/entry/show?entryid=b91ea195-fd3d-4171-bae4-198c46575c16 | PANGAEA Portal https://doi.pangaea.de/10.1594/PANGAEA.933280 | IEE Dataport https://ieee-dataport.org/documents/nenu-mpf#files |
| Reference | Rösel et al., 2012; Rösel et al., 2015 | Istomina, et al., 2020a | Istomina, 2020 | Lee et al., 2020; Lee and Stroeve, 2021 | Ding et al., 2020a, Ding et al., 2020b | Peng et al., 2022 |

One common aspect to these MPF datasets, are their spatiotemporal continuity and temporal span which still need improvement, partly due to cloud obscuration. Among these datasets, the MERIS-Istomina MPF (which is an improved version of an earlier dataset version generated in 2015 (Istomina et al., 2015)), was considered to have the highest estimation accuracy

because it was derived using a physical-based model (Istomina et al., 2015; Zege et al., 2015). However, its short temporal

span (2002-2011), makes it unsuitable for long-term trends and evolution analysis of MPF. The MODIS-Peng MPF dataset has a longer temporal span (from 2000 to 2020) and is more spatiotemporally continuous when compared to the other datasets, representing a potential advantage in terms of representing the dynamics and evolution of the MPF of Arctic sea ice more
realistically.

### 4.4.1 MPF observations: validation and training datasets

Machine learning and deep learning models require large amounts of data to be trained and validated. In the case of Earth science applications, this data often comes from in situ measurements collected by ground-based sensors, buoys, or other
instruments. In situ data provides critical information for validating and improving machine learning models, ensuring that they accurately reflect real-world conditions (Sun et al., 2022). However, melt ponds are notoriously difficult to measure, as they are constantly changing in size and shape. This makes it challenging to collect the in situ data needed to train and validate machine learning models for melt pond detection and mapping. In Table 4 lists a summary of some of the most used datasets are listed. This set of MPF observations result from multiple melt ponds surveys and campaigns conducted in the Arctic,
coming from multiple approaches or types of collection, comprising field and aerial observations. Most are linked to Arctic field campaigns meaning that contrary, to the MPF products listed in Table 2, they are irregularly spaced in time.









**Table 4: Examples of some melt pond or MPF observations open source datasets used for training models or to run independent validations**


| Dataset name and online source | Reference | Time Range | Spatial Coverage | Spatial Resolution | Format | Preview example |
|---|---|---|---|---|---|---|
| WorldView-2 (WV2) MPF | Wright and Polashenski (2019) | 2000-2015 | Beaufort Sea: 72.0◦ N, 128.0◦ W | 0.46 and 1.84 m | GeoTiff | |
| MEDEA | Webster et al. (2015) | 1999-2014 May-August | Sites locations within 69 – 85.5 ◦N | 5 to 25 km | PNG, GeoTIFF, JPEG, ASCII, Excel | |
| NSIDC | Fetterer et al. (2008) | 1999 - 2000 | 72.8–85.1◦N | 1 x 1 m | PNG, GeoTIFF, JPEG, ASCII, Excel | |
| TransArc | Nicolaus et al. (2012) | August to October 2011 | 83.1–86.3◦N | 50m~1km | csv | n.a. |
| PANGAEA | Niehaus et al. (2022) | 2017 to 2021 | Up to 82.3◦N | 10 x 10 m | NetCDF | |
| HOTRAX | Perovich et al. (2009) | Aug-Sep 2005 | 74.4–86.1◦N | 57 x 70 m | Jpeg, PNG | |
| NPI | Divine et al. (2015) | July to August 2012 | 81.4–82.7◦N | 60 40 m | Tabular | n.a. |




| IceWatch | IceWatch | 2012-onwards | Depends on the cruise | n.a. | jpeg | 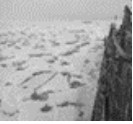 |

Some of these MPF observations, have been used as training data for deep learning algorithms (e.g. Ding et al., 2019; Ding et al., 2020b) or served as independent validation (e.g. Peng et al., 2022), as listed in Table 1. WV2, MEDEA, NSIDC are 970 PANGAEA based on high resolution satellites, they are irregularly spaced in time, while the remaining datasets are based on aerial photography (helicopter or drone-borne imagery) e.g. HOTRAX and NPI and in situ ship-based photographs e.g. IceWatch datasets.

- The NSIDC dataset, by the National Snow and Ice Data Center (NSIDC) is based on high resolution imagery classified 975 following Fetterer et al. (2008), and corresponds to the site location of the SHEBA experiment (Beaufort Sea, East Siberian Sea, Canadian Arctic and Fram Strait). The data can be accessed at https://nsidc.org/data/g02159/versions/1#anchor-2.

- The MEDEA (Melt Pond Fraction Statistics From High Resolution Satellite) Images, are retrieved by the Polar Science Center, University of Washington, also based on visible bands of high-resolution optical data (1 m resolution). 980 The statistics of melt pond coverages were retrieved at locations in the Beaufort Sea, Chukchi Sea, the Canadian Arctic, the Fram Strait, and the East Siberian Sea from May to August, following the methodology of Webster et al. (2015). The data can be accessed at http://psc.apl.uw.edu/melt-pond-data/.

- The WV2 MPF dataset, based on WorldView satellite imagery, results from being processed using the Open Source Sea-Ice Processing algorithm developed by Wright and Polashenski (2018). The data can be accessed at 985 https://arcticdata.io/catalog/view/doi:10.18739/A22Z12P4J.

- The PANGAEA dataset is based on 31 scenes of cloud-free Sentinel-2 data by Niehaus et al. (2022). A special focus was given to cover the Multidisciplinary Drifting Observatory for the Study of Arctic Climate (MOSAiC), in fact all scenes from 2020 cover the expedition path. The data can be accessed at https://doi.pangaea.de/10.1594/PANGAEA.950885?format=html#download.

- The HOTRAX MPF dataset were collected during the Healy Oden Trans-Arctic Expedition (HOTRAX) by the Polar Science Center, University of Washington (Perovich et al., 2009). The dataset contains mapped melt pond zones through deep learning approaches (Sudakow et al., 2022) from aerial photographs obtained during helicopter photography flights, as part of HOTRAX campaign. The data can be found at https://zenodo.org/record/6602409#.ZFuxlHZBzmE.





- The NPI data were collected by the Norwegian Polar Institute (NPI) during the ICE12 field campaign on Arctic sea ice north of Svalbard during the summer of 2022.The dataset comprises of fractions of three surface types (bare ice, melt ponds and open water) from which MPF were calculated (Divine et al., 2015), along the flight tracks calculated from images collected by a helicopter-borne camera system. The data can be accessed at https://data.npolar.no/dataset/5de6b1e4-b62f-4bd4-889c-8eb7bb862d3b.

- The TransArc melt pond observations were collected from the ice breaker RV Polarstern during the Germany Trans-Polar cruise ARK-XXVI/3 (Nicolaus et al., 2012), through hourly observations from the bridge of the research vessel on 29 August and 6 September. The data can be accessed at https://doi.pangaea.de/10.1594/PANGAEA.803312.

- The IceWatch observation of MPF are collected and made available through the IceWatch community. IceWatch is a program that coordinates visual observations of the sea ice conducted from ships in the northern hemisphere, using
a tool named ASSIST, which is a web-based application in which one can record visual observations of sea ice, and can be run at sea without an internet connection. Both ASSIST and the MPF by the community can be accessed at https://icewatch.met.no/assist.

Worthy of a mention are also the DLUT data (Lu et al., 2010; Huang et al., 2016) from helicopter collections, and the ship-
based datasets JOIS (Tanaka et al., 2016) and PRIC-Lei (Lei et al., 2017). These three datasets are not available online (and hence have not been included on Table 4), but are nevertheless important MPF observation-based datasets that have been used in several studies, since they have been provided upon request directly to the first authors. DLUT MPF observations were collected during two Chinese Arctic Research Expeditions by the Dalian University of Technology (DLUT), in 2008 and 2010, leading to over 9000 images. These images were then classified into three surface types (sea ice/snow; water and melt ponds)
(Huang et al., 2016). JOIS MPF were collected from ship-based observations by Joint Ocean Ice Study during 2003-2014 on the Canadian Coast Guard Ship Louis St-Laurent (Tanaka et al., 2016), with a spatial resolution 1453 m2~2397 m2 and a spatial range of 68.9–88.2N, the images, collected by a camera mounted with a view of the horizon and ice pack where classified into 5 classes (water, ice, water and ice, pond and ice, water, pond and ice). Finally, PRIC-Lei melt pond observations were collected during the Arctic Research Expeditions by the Polar Research Institute of China (PROC) during summer from
2010 to 2016 (Lei et al., 2017), where melt ponds, along other variables, such as snow thickness, sea ice concentration, were documented half-hourly from the bridge of the R/V Xuelong, with a coverage of 1 x 1 km measured along the cruise at 71.7-88.4∘N.

Despite the large area covered of the Arctic Ocean by the MPF observation data collections, the challenges and high costs
associated with expeditions in the high Arctic Ocean result from a lack of uniformity in their temporal distribution, thereby exhibiting irregular spacing across time intervals or spatial coverage. Furthermore, there is only limited data available for recent years, offering an extra challenge for the validation of training of satellite missions launched after these collections. As a result, there is currently a significant shortage of high-quality data for this important area of research, especially





considering the need for a considerable amount of training datasets required by data-hungry models such as deep learning Sun
et al., 2022.

**4.5 Radar-based limitations to MPF retrievals**

The major advantage of SAR in respect to optical and thermal IR sensor-based methods, is that SAR is weather independent making it more suitable than the other sensors – as these are often of limited use given predominantly cloudy seasons
characterising the Arctic region. SAR provides thus a less limited, greater temporal coverage in opposition to optical sensors. Backscattering coefficient ($\sigma\circ$) is considered a key feature in MPF estimations (Han et al., 2016; Fors et al., 2017), since the lack of $\sigma\circ$ intensity of melt pond surfaces compared to the sea ice could potentially be used for MPF retrieval using $\sigma\circ$, since backscatter intensity becomes weaker with MPF (Han et al., 2016; Fors et al., 2017). However, despite microwave frequency sensitivity to pond fraction (Howell et al., 2020), caution should be taken in regards to SAR ability to make estimations of the
cover of melt ponds, due to complexities introduced when ponds are subject to wind roughening (Yackel and Barber, 2000).

As concluded by Fors et al. (2017), intermediate wind speed (i.e. around 6.3m/s) allows for backscatter from the melt ponds using the R(VV/HH). In addition, Scharien et al. (2014b) concluded that wind speeds above 5m/s reduced the correlation between R(VV/HH) and MPF, leaving a very narrow wind speed interval for melt pond retrieval with X-band SAR. For example, while Fors et al. (2017) and Scharien et al. (2012, 2014b) found strong correlation between R(VV/HH) and MPF in
low-wind cases, Scharien et al (2014a) found R(VV/HH) to increase with MF and low-wind speed (1.1m/s). The weak melt pond backscatter, combined with a low signal-to noise ratio, hamper the use of difference in polarimetric properties between sea ice and melt ponds for melt pond fraction retrieval Fors et al. (2017). For low-wind scenarios with X-band, backscatter intensity becomes weaker with MPF (Han et al., 2016; Fors et al., 2017) in contradiction with in C-band Yackel and Barber (2000) which found no correlation between $\sigma\circ$ with HH polarization and MPF also under calm wind conditions (1.5m/s). In
their study, C-band SAR also shown to have a significant positive linear relationship between $\sigma\circ$ and melt pond coverage but this became weaker under moderate wind speeds.

Another considerable constraint of SAR for melt pond studies or MPF retrievals is the fact that the radar backscatter changes with the incidence angle, meaning that the same pond may have different signatures at different angles. Incidence angles above ~40°, restricted the use of X-band in MPF retrieval (Fors et al., 2017), while 29° showed to provide the best results, suggesting
that future studies should focus on incidence angles in the range of 29 to 40°. On the other hand, the best incidence angles to monitor melt ponds on Arctic multiyear sea ice using TerraSAR-X dual-polarization data were in the range of 20-30 degrees in the study of  Han et al. (2016). This study found that at these incidence angles, the backscatter from the melt ponds is relatively strong, while the backscattering from the surrounding ice was relatively weak, making it easier to distinguish melt ponds from the surrounding ice. The study also found that the use of dual-polarization data can further enhance the detection of melt ponds, particularly in areas with thin ice. In C-band studies, backscatter from melt ponds is relatively weak at lower





incidence angles (20 to 25°) (Yackel and Barber, 2000) while at greater incidence angles (~30°), σ° observed a significant increase, being these attributed to the specular reflection of the SAR signal and increased roughness of melt ponds at higher incidence angles, respectively. It should be noted that the optimal incidence angle for monitoring melt ponds may depend on factors such as the type and condition of the ice, as well as the characteristics of the imaging system. Also, constructing a
mosaic based on SAR imagery over large regions presents challenges thanks to the incidence angle variability as pointed out by Howell et al. (2020), which is another constraint to generating Arctic wide radar-based MPF products.

In addition to the above-mentioned constraints faced by radar-based MPF and melt pond detections, the consideration of polarizations and frequencies is important (e.g. Yackel and Barber, 2000; Fors et al., 2017). Most studies using SAR faced limitations in retrieving melt ponds since the σ° and polarization ratios were not enough to discriminate melt ponds from sea
ice and open water, leading to cases of under or overestimation. In some cases overestimation might be due to the assumptions made, as for example, in the study from Li et al. (2017) who got systematic overestimations and dependent deviations, which might due to (1) melt ponds having the same polarimetric characteristics as the open ocean water and (2) FYI being assumed as bare ice, ignoring the effects of snow, snow depth, ice deformation and wet snow cover on the ice.

Furthermore, the resolution of the data may not be sufficient to accurately detect and monitor small ponds. Additionally, the
temporal resolution of the data may not be sufficient to capture the dynamic changes in the pond coverage over short time scales. Finally, the presence of shadows or other objects in the scene can result in false detections or misclassifications (Han et al., 2016). Another important consideration is that most studies have focused on FYI. Melt ponds on MYI exhibit different physical and microwave scattering characteristics from those on FYI. Since polarimetric features from melt ponds on MYI are likely different from those on FYI, further examination and evaluation of melt pond retrievals from Polarimetric SAR data
should be also conducted for MYI. Finally, to date there is a lack of extensive studies comparing surface roughness and incidence angles for different SAR frequencies and polarizations, and over different sea ice surface types, as much efforts have been made towards single polarization HH (Scharien et al., 2017; Howell et al., 2020) and mostly focused on RADARSAT (Li et al., 2017; Ramjan et al., 2018; Howell et al., 2020) and over FYI. Therefore, these results may not be applicable in all cases, and further research is needed to generalize the findings.

**5 Identification of challenges and opportunities**

Despite the pioneering work done on morphological properties of melt ponds and their distribution using in situ and satellite-based sensors (e.g. Yackel and Barber, 2000; Eicken et al., 2004), there is still a lack of full understanding and proper modelling of melt ponds seasonal evolution coverage over different sea ice conditions, as well as of their spatio-temporal changes in depths and coverages across the Arctic region (Webster et al., 2015). The importance of melt ponds and MPF knowledge goes
beyond their hindering impact on the understanding of the energy budget in the Arctic region. The following listed key aspects



show areas where there is a poor knowledge of melt pond that is impactful and/or where there is potential to harness and overcome those gaps.

**5.1 Melt pond parameterization and climate models discrepancies**

The behaviour of melt ponds is pointed as a main source of uncertainty in climate models (Flocco et al., 2012; Webster et al., 2015), due to insufficient physics knowledge on melt pond evolution and the inability of climate models to resolve such high-resolution features. Melt ponds were typically represented in global climate models simply through the adjustment of the overall sea ice albedo in the summer months, which is insufficient since it does not account for their variability with respect to ice type, sea ice topography or the albedo itself. Incorporation of melt pond observation processes into model simulations have yielded conflicting results for the trend in melt pond fraction: Schröder et al. (2014) found an increase in MPF, whereas Zhang et al. (2018) found no statistically significant trends in MPF per unit ice area. Melt pond parameterizations vary greatly by model (Scott and Feltham, 2010; Hunke et al., 2013; Zhang et al., 2018), causing different effects on simulated sea ice and discrepancies in model simulation of long-term changes in melt pond fractions. In some model simulations, sea ice is highly sensitive to melt pond drainage parameters (Scott and Feltham, 2010), while in others, sea ice is strongly affected by the sequence in which ponds from across ice thickness categories (Flocco et al., 2010).

The advances in melt pond parameterization have improved their representation in sea ice and coupled ice-ocean models (Flocco et al., 2010; Flocco et al., 2012; Schröder et al., 2014; Zhang et al., 2018) and it has been shown that sea ice model predictions were sensitive to melt ponds inclusion (Flocco et al., 2012). Melt pond processes are thus one example of incomplete model physics that may contribute to the discrepancies in the rate of Arctic sea ice loss between climate models (e.g., Stroeve et al., 2012). The diverging results earlier described suggest the need for reducing uncertainties in melt pond parameterization in order to advance the knowledge of seasonal evolution of melt pond coverage over different sea ice conditions and spatio-temporal change in pond depth and coverage.

Another crucial aspect that a better knowledge of melt ponds could result is the inclusion of light reflection by melt ponds into climate models is of great importance (Flocco et al., 2012), especially in times witnessing strong changes (Zege et al., 2015, Tschudi et al., 2008; Rösel et al., 2012). As the melt pond fraction is closely connected to the relief of the sea ice (Polashenski et al., 2012), the maximum pond fractions is expected to increase as well. Therefore, including light reflection by melt ponds into climate models is an important task, particularly in light of the environmental changes observed recently (e.g. Flocco et al., 2010, 2012; Hunke et al., 2013). A physical model of the reflective properties of melt ponds is needed for understanding the physics of sea ice, as well as for the accurate interpretation of the results of remote sensing and field measurements (Tschudi et al., 2008; Rösel et al., 2012; Zege et al., 2015).




### 5.2 Impacts of melt pond on ice concentration retrievals

Melt ponds are also responsible for resulting in  less accurate sea ice concentrations (SIC) derived from satellite-based microwave brightness temperatures during the summer (Cavalieri et al., 1990; Kern et al., 2016; Kern et al., 2020). This occurs because ice concentration retrievals from microwave radiometers are degraded by the existence of melt ponds during the melting season (Cavalieri et al., 1990; Kern et al., 2016). Melt ponds are opaque and can  contaminate passive sensors such as AMSR-E, and have been reported to underestimate the sea ice concentration during the summer (Comiso and Kwok, 1996; Mäkynen et al., 2014; Kern et al., 2016). This is another reason to reinforce the need of datasets in the area of MPF, in order to allow the development of ice concentration retrieval algorithms, for the quantification of their uncertainties and to optimize overall sea ice concentration techniques (Kern et al., 2016; Kern et al., 2020). Furthermore, attempts of relating MPF and SIC from microwave data have also proven  unsuccessful (Fetterer et al., 2008).

### 5.3 MPF potential to improve sea ice extension predictions

The knowledge of melt pond spatial distribution and the length of melting season is relevant to predict the role of sea ice cover in the radiative balance of the region (Schröder et al., 2014; Liu et al., 2015, Ding et al., 2019). To this end, a much larger number of observations of the MPF for different sea ice types at regional scale are needed in order to improve our understanding of the role of melt ponds in the Arctic climate system. In fact, recent studies suggested that the knowledge of MPF over Arctic sea ice can be used to improve the predictions of summer Arctic sea ice (Liu et al., 2015; Schröder et al., 2014; Ding et al., 2019; Howell et al., 2020). Thus, accurate knowledge of melt pond variation in the Arctic is of great importance for the understanding and prediction of sea ice changes (Schröder et al., 2014; Liu et al., 2015; Howell et al., 2020). MYI and land-fast FYI have been the main focus in previous studies, highlighting the need of expanding these retrievals efforts to other sea ice types which are shown to become more prominent (like drifting FYI), as the Arctic is shifting to a thinner and mobile sea ice cover. Despite the strong correlation found between MPF in May-June to sea ice extent in September, little work has been done to elucidate the linkage between melt ponds from May-July and the extent of Arctic sea ice. An additional interest on MPF knowledge comes from the fact that in recent years it has been shown that the MPF in spring can deliver predictions of September minimum sea ice extent (Scharien et al., 2017; Liu et al., 2015; Ding et al., 2019; Ding et al., 2020a), which leads to enhancements in the seasonal forecasting ability of the summer sea ice extent. On the other hand, Liu et al. (2017) found no relationship between MPF and September sea ice.

### 5.4 Sensor-based limitations and improvement of radar-based approaches

The methods and sensors also offer a set of challenges to get better knowledge of melt ponds. While optical is the most traditional, it is  heavily constrained by cloud cover (e.g., Fetterer et al., 2008; Howell et al., 2006), however it is still a challenging task to retrieve melt pond coverage from SAR images at larger scale (Huang et al., 2016). To date, there is no solid



SAR-based product when compared to aerial or optical data, being MYI of particular lack of strong results (Yackel and Barber, 2000; Mäkynen et al., 2014; ; Landy et al., 2014; Tanaka et al., 2016; Li et al., 2017). Additionally, intra-daily variations of the melt pond fractions due to the strong diurnal cycle of melt-water production rates are observed at least during the first part of the melt season (Eicken et al., 2004). Therefore, satellite and surface based observations strongly depend on the point in

time of the observation and discrepancies in the melt pond fractions of both are not surprising. Despite present-day high-resolution SAR sensors (e.g. TerraSAR-X, COSMO-SkyMed, RADARSAT-2, Sentinel-1) which can achieve up to 10 and 1-m spatial resolution, SAR data has not been fully explored in mapping melt ponds at a large scale - which can be due to the previous spatial SAR resolution and speckle noise issues, along with all surface roughness and radar parameters constraints.

Regarding polarization-related challenges, the first studies with SAR had limitations in the retrieval of melt ponds (e.g. Yackel and Barber, 2000; Kim et al., 2013; Mäkynen et al., 2014), since they used only single-polarization (single-pol) backscatter information. Single-pol coefficients depend mostly on surface roughness. Hence, melt ponds which have smooth characteristics had limited information to be identified. Co-polarization (co-pol) ratio, i.e., the ratio of backscattering derived from VV and HH polarization has been used to identify melt ponds on FYI  (e.g. Fors et al., 2017) thanks to existing contrast

of dielectric properties between melt ponds and surrounding ice (Scharien et al., 2012). Furthermore Scharien et al. (2014b) concluded that co-pol ratio holds the potential for unambiguous detection of FYI melt pond formation, as it is distinct from the background snow-covered or bare ice, as long as ponds are in liquid state. Studies on summer MYI have also been addressed with the co-pol ratio method (with TerraSAR-X) dual co-pol data (Han et al., 2016). More recently, it has been shown that multiple-polarization (multiple-pol) radar can improve melt pond observations and monitoring (Fors et al., 2017). And finally,

also for the retrieval of MPF, the use of single-polarimetric SAR has proven to be difficult. The main achievements on MPF retrieval with SAR came from dual-polarimetric C-band on FYI. R(VV/HH) was found to be the most promising SAR feature (Fors et al., 2017) however, for the case of X-band wind is more limiting than in C-band (for medium-wind conditions, such >5m/s). Late studies revealed the potential of MPF estimation from X-band SAR, highlighting the importance of including wind speed and incidence angle to ensure a more robust MPF retrieval algorithm (Fors et al., 2017).

Polarimetric parameters were not enough to discriminate melt ponds from open water. Spatial texture (from average and standard deviation) of polarimetric features improved the results (Han et al., 2016). Future approaches with SAR should (1) include multi-polarization SAR in different angles for various ice types, (2) study the relationship between the polarimetric signatures and microwave emissivity of melt ponds and (3) monitor melt pond fractions over the entire Arctic using the ensemble of multi-polarization SAR data and passive microwave observations.

**5.5 Area and ice-type coverage**

Melt ponds' knowledge has been traditionally obtained from land-fast ice observation (Yackel and Barber, 2000; Perovich et al., 2002; Eicken et al., 2004; Polashenski et al., 2012), thanks to the relative ease of revisiting the same study area. This is not the case for locations of difficult access making them both costly and dangerous to reach, and presenting other types of sea

ice. For instance, given its importance in the Arctic context, the melt ponds on first-year sea ice in the Canadian Arctic

Archipelago (CAA) have been investigated in several studies (e.g. Yackel and Barber, 2000; Scharien and Yackel, 2005), and even though there has been already the effort on studies encompassing both MYI and FYY (e.g. Scharien et al., 2007; Wright and Polashenski, 2018; Buckley et al., 2020; Li et al., 2020) there are still few studies investigating melt ponds on both ice types. In fact, differences in melt ponds over different sea ice types, in terms of depths, sizes but also evolution throughout the melting season are still a source of ambiguity which may be related to biases and uncertainties in retrievals (Wright and

Polashenski, 2020). Further research is needed to generalise conclusions and parameters across different ice types, for both types of sensors. As already discussed, polarimetric characteristics of melt ponds vary according to the ice type where they develop, but also optical-based studies revealed different results according to sun elevation and surface type (e.g. Rösel and Kaleschke, 2011).

Furthermore there are the existing constraints regarding the coverage of the Arctic, given the fact that missions are not covering up to 90 degrees North. For instance, although MODIS has a maximum latitude coverage of approximately 85 degrees, with some variations depending on the specific product or dataset being used, MODIS still can be used to generate sea ice data products for the entire Arctic region, including the North Pole. This is because MODIS is able to capture imagery of the Arctic region from various angles and perspectives, which can be used to create a composite view of the sea ice coverage. To extend

data coverage to the North Pole region (90 degrees north), data assimilation techniques are often used to allow the combination of data from multiple sources (including satellite data, ground observations and numerical models). The same shortage of data applies to in situ (ship-borne or airborne) MPF observations, given associated costs and challenges of conducting expeditions in the high Arctic.


## 5.6 Standardisation of ice and melt pond types

The lack of standardised surface type definitions also adds up to the challenge of quantitatively monitoring sea ice

characteristics (Wright and Polashenski, 2008). One of the key tasks in creating training datasets for machine learning techniques is to have expert human classifiers who will train the algorithm according to definitions of each surface type that are broadly agreed by the community. While the definition of open water, ice and melt pond might at first seem a straightforward process, there are very different approaches within the cryosphere community when it comes to defining classes or types of ice surface. For instance, studies focused on the classification of melt ponds do not share a common approach

on class definition (Webster et al., 2015; Huang et al., 2016; Wright and Polashenski, 2018; Buckley et al., 2020). Also, the skill of machine learning prediction increases substantially as the size of the training set grows, however the creation of training



data is a task that consists of manual classification by humans. The manual classification of these datasets although time consuming would profit from experts and would accelerate the uptake of AI-approach to melt ponds studies.

## 6. Conclusions


Despite melt pond's relevance to the Arctic's energy budget, given their role in reducing surface albedo, current global climate and sea ice models do not model melt ponds on sea ice. In fact, limited observations of melt ponds are far from adequate for the Arctic Ocean which results in a lack of spatial and temporal scale-knowledge information and uncertainty of these water bodies. The poor understanding of the physics governing pond evolution and the resulting lack of realistic parameterizations,

are pointed as the main source of uncertainty in climate and sea ice models predictions. Furthermore, the high temporal and spatial change in melt ponds, hinders estimations of Pan-Arctic scale changes based solely on field observations.

While presently long- term distributions from melt ponds are substantially deduced from models, only large-scale and space-based observations can truly show the changes in Arctic melt pond fraction (MPF) – an important geophysical parameter used in the context of climate and sea ice research. Optical imagery has traditionally been the most widely used approach to estimate

large-scale MPF from space, followed by Synthetic Aperture Radar (SAR) and passive microwave data. Despite its wider usage, optical data presents considerable drawbacks, such as (1) cloud cover which limits access to information; (2) conventional MPF algorithms treat melt ponds as features with constant reflectance – which is inaccurate and leads to lack of representativeness (since the spectral reflectance of ponds vary greatly) and (3) the low resolution used for optical-based studies of meltdowns disregard smaller ponds inducing under- and overestimation results. These limitations of optical hamper

satisfactory information and reliable MPF products. On the other hand, radar-based approaches, namely Synthetic Aperture Radar (SAR) offer a unique opportunity for retrieving MPF given they are weather and illumination independent. Research has been developed using different polarization combinations and although promising results in predicting MPF ahead of its formation in spring, there are still significant obstacles such as (1) unavailability of cross-polarization and multi frequency studies retrievals, (2) wind speed information and associated surface roughness; along with (3) different ice types and incidence

angles which contribute to deviating findings causing discrepancies or agreement on MPF evolution across different ice types and correlation with other variables. These challenges result in that it is still not possible to generalise results and to date there is no solid SAR-based product compared to aerial or optical data. Finally, (1) passive microwave sensor's brightness temperature is highly sensitive to atmospheric conditions and (2) this type of sensor has normally lower spatial resolution compared to optical and SAR sensors, which limits their applicability to MPF retrieval.

Recent advances in algorithms for MPF retrieval, using AI, namely machine learning, and more recently deep learning algorithms have shown improved results compared to conventional approaches. Besides accelerating an otherwise time consuming process, AI also holds the potential to depict complex patterns and spatio-temporal evolution and correlation.



Moreover, since deep learning algorithms learn from data and perform feature extraction without the need of human labour, it compensates for the current lack of existing training datasets for such specific sea ice features in inaccessible locations,

underscoring their probable significance in future research. With the prevailing sea ice type shifting from multi-year ice to first year ice, where melt pond coverage is wider, emphasizing the urgency to obtain reliable, improved accuracy and a wider coverage of MPF data to study these processes. As a result, continuing to improve melt pond observations and summer melt process studies across the Arctic is critical, taking advantage of existing microwave sensors on pan-Arctic scale and applying new algorithms is a critical step to better understand their impacts and support understanding of Arctic sea ice and Arctic

climate system.










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
