# Peer review of "Review Article: Earth observations of Melt Ponds on Sea Ice"

_The Cryosphere, 2023_

## Referee Comment (RC1)

Review on tc-2023-75, **Review Article: Earth Observations of Melt Ponds on Sea ice**

The manuscript is dedicated to the sea ice-covered Arctic Ocean during melting season and comprises a literature overview of the currently published melt pond retrievals and datasets. It lists the published retrievals and datasets, including optical of very high to moderate resolution, passive microwave (PM) of various frequencies, and synthetic aperture radar (SAR)-based methods, as well as the datasets available publicly online. The overview is comprehensive, and the topic is certainly important as melt ponds are currently one of the major uncertainties in global climate models and also regional ocean-atmosphere coupled models of the Arctic Ocean, which is important in the context of the recent change and Arctic amplification.

As the topic is of value to the scientific community, and the amount of work and research invested into the manuscript is substantial, it is even more important to address possible unclarities and unfortunate formulations at the current stage so that the reader gets the most added value out of the final version of the manuscript. Therefore, the reviewer suggests several modifications and improvements on this overview manuscript before it can be considered for the publication in the Cryosphere.

The reviewer comments are listed below, first as general remarks, and then as specific comments. Please note that some of the specific comments are by no means just technical corrections but major concerns.

(1) In general case, as a typical review paper contains no own scientific research, the added value comes from the neutral and expert-level evaluation of a given scientific field. In this manuscript, unfortunately, this does not hold. There are some faulty statements which go against the first principles, which is unacceptable and must please be addressed by the author in particular detail – examples in the specific comments below.

(2) Different amount of attention is given to different publications with clear polarization of the author towards SAR and machine learning approaches. The reviewer is not convinced from the text that this is the objective reality of the scientific field. Challenges of all the discussed approaches need to also be addressed equally, there are certainly also critical challenges of the AI approaches like overtraining and unreliability in the times of change – as is currently the case in the Arctic. This needs to be addressed.

(3) The text is not concise, lots of repetitions, contradictions, which renders the manuscript unusable to the reader, with the exceptions of the nicely brought together lists of the products and retrievals as Tables 1-3. Instead of repeating a single reference many times in one paragraph, (e.g. Scharien et al 2012 on P14 L360-367, or Zege et al 2015 on P20 L547-555, and throughout the text), better condense those paragraphs so that you only include a given reference once. Also, the point you are making will be clearer to the reader, which is not the case now.

(4) The usage of the references needs to be addressed. The manuscript gives the impression that the separate sentences have been extracted out of the corresponding reference papers and put together in a somewhat loose order, resulting in contradictions or repetitions in the text. Errors in references, to the point of not being able to follow how a given reference ended up after a given sentence – very misleading for the reader.

(5) The manuscript is unnecessary long. A lot of volume is spent on theoretical background. It is better to refer the reader to published work if needed, no need to write it out here. The text needs to be condensed before it can be helpful to the reader. Some specific suggestions as to how are in the comments below.

**Recommendation:**

Currently, the manuscript looks more like a thesis chapter, so that the format does not fit a Cryosphere article. It would fit better if reduced and condensed at least to half the size, leaving only the content which makes impact. Together with own reconsideration as per reviewer comments, maybe some coauthors could be invited, ideally with a long-term expertise in the field of melt pond in situ and/or remote sensing. Then the manuscript can be reviewed and reorganized so that as end result the reader could profit from an objective expert-level overview. The reviewer encourages the author to invest this additional time and effort.

The reviewer recommends a major revision, better withdraw and resubmit when ready, at the discretion of the editor.

**Specific comments to the text (Page/Line):**

P1 L 10 and later: "wider melt pond coverage" – unusual term to use, more frequent would be "higher melt pond coverage"

P 1 L 15: "However, observations of melt ponds are far from adequate for the Arctic Ocean…" – I would reformulate this sentence towards something like "However, the coverage of available in situ data is limited for both FYI and MYI and satellite remote sensing is be used to obtain a better data coverage."

P1 L 17: "This paper provides an overview of efforts in EO remote sensing studies of melt ponds, for both optical and radar-based approaches." – there needs to be clarity as to which methods and products you discuss in the paper, as in: in situ, airborne, ship-borne, satellite of which wavelength and resolution. I think this point of the abstract is a good place to list them all so that the reader can decide whether the paper is of relevance to him/her.

P1 L 19: "ranging from the early traditional techniques to the increasingly prevalent use of Artificial intelligence (AI)"– Ambiguous term, unclear what is meant under "early traditional" techniques. Could be threshold-based or spectral unmixing or neural networks etc, please specify for clarity.

P1 L 20: "The current large-scale optical-based pan-Arctic MPF datasets are intercompared along with the main advantages and disadvantages" - you write you intercompare the current large-scale optical MPF datasets, and this is something the reader actually hopes to see. However, I do not see a scatter plot or a critical discussion with a comparison of the quality of the datasets, I only see Table 2 which is a list, not a comparison. If you strive to include an intercomparison, you need to include a scatter plot of the products, where available, or e.g. an example timeseries one summer. Otherwise this sentence needs to be reformulated.

P2 L 48: "Albedo (α) (or spectral reflectance), that can be defined as the ratio… "- albedo is not spectral reflectance. Here, but also further in the manuscript there is an impression that the author does not distinguish between albedo (the ratio of two hemispheric integrated values) versus reflectance (ratio of directional or angular value to directional or angular or hemispheric) – even when using the MODIS top-of-atmosphere reflectance later in the paper (P 35 L 820), which is obviously directional as it is measured by a satellite sensor. Or the same with spectral and broadband albedo, e.g. on P6 L 152 – do you mean spectral or broadband albedo? Please take care to specify the exact term each time and check e.g. Nicodemus 1977 for a convention on albedo and reflectance values.

P2 L 54: "Ice albedo's temporal and spatial variation is closely related to the global climate change and regional weather system due to the above mentioned feedback mechanism (Henderson-Sellers

and Wilson, 1983)" – this reference here is extremely confusing as the feedback mechanism you refer to here was introduced 12 years later by Curry et al 1995, as you state two lines earlier (P2 L 52). Please reformulate.

P 5 L 105: "Figure 2 shows also an example of the albedo spectral dependence for white ice, snow and two types of melt ponds - light and dark." – I do not see the light pond in Figure 2. It stands "melting ice" where it needs to be "light blue pond". Generally, there is no need to include a figure from Grenfell and Maykut (1977) today. Please check out in situ spectral measurements of sea ice and melt ponds at PANAGEA, either with RAMSES or with ASD spectrometer. Also, Fig. 2 represents each surface type as just one curve. It might be useful to state that these are just sample curves and surface variability is in fact much greater.

P 5 L 135: "Melt ponds start to form in the Arctic at the end of May, covering a considerable portion of the sea ice surface by mid-June" – compare to P 7 L 167: "The ice-ponded coverage is highly variable going from zero percent in mid-June…" – ambiguous, is it "zero" or "considerable portion" in mid-June? Your source, Fetterer and Untersteiner, 1998, was published in the times of a very different Arctic 25 years ago. Maybe you could consult one or several of the modern MPF products that you cite for this instead.

P7 L 189 "thick multiyear sea ice MYI has been replaced by thinner first-year ice (Kwok, 2018) versus P8 L 194 In addition, the prevailing sea ice type has been changing from MYI to FYI in recent decades (Comiso, 2012)" – there is no need to repeat same thing several times on many pages with different references. Also, you have already used the abbreviation MYI before P7 L 189, no need to introduce it again here.

P9 L 18-20 vs P9 L36-39 – please merge these repetitions.

P9 L 246 – please specify which radar you mean here, also I think you mean passive microwave sensors here as well as you mention them further on in the text.

P 13 L310:" … enabling the use of increases in brightness temperature as an indicator for the presence of melt over snow and ice (Mote et al., 1993)" – The Tb_h of water and melt pond is lower than that of dry sea ice or snow, and the presence of melt ponds on top of sea ice in fact lowers the brightness temperature, see Fig. below. Please reformulate so that there is no confusion.

[Figure]

(Figure source: Spreen et al., 2008)

P13 L313: "since it is the frequency that has the lowest brightness temperature over dry firn, which maximises the change in brightness temperature caused by the emergence of liquid water (Johnson et al., 2020)" – since your paper is about sea ice and not glacier or land ice, firn has no relevance here and just adds to further confusion. Please remove this link and reformulate the sentence.

P9 L 316: "resulting in a major shortcoming of today's suit of SIC algorithms (Rösel et al., 2012; Kern et al., 2016)". – it is ambiguous whether you refer to the shortcomings of the PM SIC here, or of the optical SIC e.g. in Rösel et al., 2012. Both have similar problems of mixing up melt ponds and open water. I think in this case you mean the PM SIC algorithms in the presence of summer melt. In this case, please look up and refer to:

Ivanova, N., Pedersen, L. T., Tonboe, R. T., Kern, S., Heygster, G., Lavergne, T., Sørensen, A., Saldo, R., Dybkjær, G., Brucker, L., and Shokr, M.: Inter-comparison and evaluation of sea ice algorithms: towards further identification of challenges and optimal approach using passive microwave observations, The Cryosphere, 9, 1797–1817, https://doi.org/10.5194/tc-9-1797-2015, 2015.

P 13 L 323: "strong sensitivity to micro-sale surface roughness" – micro-scale

P 13 L 324: "which leads reduced volume scattering"– leads to reduced volume scattering…

P14 The entire page 14 has nothing to do with melt ponds. I suggest to reduce Section 3.1.1 only to a couple of definitions if absolutely necessary, and refer the reader to key publications for more radar theory.

P15 L377: "Under calm window conditions" – under calm wind conditions

P 15 L 395: "Moreover, as melt ponds presence on sea ice affect its physical and dielectric properties, it leads to increased penetration of the radar signal into the ice, making it difficult to accurately measure the thickness of sea ice using radar during periods of advanced melt (Scharien et al., 2010)." – the presence of melt water leads to increased absorption of the radar signal, not its penetration. Also, Scharien et al 2010 does not talk at all about sea ice thickness measurements using radar. Please check.

P 16, Section 3.2 Optical and SAR technical abilities for melt pond discrimination – you probably want to include also passive microwave here for consistency, I, however, would merge the entire remote sensing background subsections (entire Section 3) into one short section.

P 16 L 404: "and for surface energy balance studies (Istomina et al., 2016; Lu et al., 2018)" – Istomina et al 2016 and Lu et al 2018 do not deal with surface energy balance. Please check.

P16 L 427-428: "the fact that since MODIS pixels used in their study had a much lower resolution, this could tend to over/underestimate melt pond coverage when a large/small number of melt ponds occupy a single MODIS pixel (Lee et al., 2020). Coarser resolution of optical data, can result in unrealistically large melt pond fraction…" - it is not evident to me how, and also contradicts to your Eq.3 on P19, where there is no pixel size in the MPF definition. Please reformulate for clarity.

P18, the caption of Fig. 7: "Figure 7: Overview of Earth observations (EO) studies of melt ponds and MPF retrievals since 2000 with highlights to EO platform (air- or spaceborne), sensor and bands used.4 Melt pond fraction (MPF) satellite-based retrievals" - please remove the section title out of the figure caption.

P19 L 510 "(or RMSE, a common measure of the difference between values predicted and the actual observed values) of ~10%, 4-12% and 45% (Zege et al., 2015; Rösel et al., 2011 and Istomina et al., 2016," – No need to write out what RMSE is. Also please check the years of references. Rösel et al

2011 or rather 2012? Istomina et al 2016 or rather 2015? Also, in this section 4 you really need to make a separation whether you talk about moderate resolution or high resolution optical sensors. They typically have different spectral resolution in addition to spatial and therefore very different approaches which should not be mixed. E.g. P19 L 512: "In order to be detected, the diameter of a melt pond should be larger than the pixel size (Wang et al., 2020)" – is obviously about very high resolution sensors which is probably irrelevant here, as you further talk about moderate resolution sensors P19 L514-515.

P19 L 516: "… are widely used in MPF retrieval … because of their strong intuitive and interpretative capability (Qin et al., 2021)." – I do not understand what a strong intuitive capability of moderate resolution optical sensors should mean. Please reformulate or remove.

P19 L521: "traditional or more recent approaches with Artificial intelligence (AI) are later summarised in Table 3." – Table 3 is the first you refer to before Table 1 and Table 2. Please rename the tables accordingly to their order of use. Also, the use of adjectives "traditional" and "recent AI approach" let the reader assume that all the work on the MPF retrievals based on physical principles is no longer happening and only AI is currently being proceeded. This is certainly not true, please reformulate for more clarity.

P20 L540: "Rösel et al. (2012) used an AI-based approach" – Rösel et al used neural network in their approach, not AI. I do not think AI usage is appropriate here, as it, although correct, still is too broad to be specific.

P20 L544: " … to understand the impacts of sea ice by the melt pond coverage" – I believe you meant "to understand the impact of melt pond coverage onto the PM sea ice concentration".

P 20 L 550: "… to simulate the bidirectional reflectance factor (BRF), black-sky and white-sky over the surfaces of melt ponds …" – Zege et al. simulate black-sky and white-sky **spectral albedo**, please include.

P 20 L 553: "For instance, melt ponds have a higher reflectivity than bare ice, especially in the visible range, due to their smoother surface and lower absorption of solar radiation." – This sentence contradicts your Fig. 2 where bare ice lies above any type of melt pond, and the model of Zege et al. 2015 and consecutive same model of Malinka et al 2016,2018 certainly do not give melt ponds brighter than bare ice. Please check.

P 21 L 564 – here is the start of the high resolution sensors, please separate is from the moderate resolution with maybe a new subsection, and the same for P22 L 592 where airborne sensors start, also further the the Huang et a 2015 at Line 605 needs to be brought to moderate resolution satellite studies.

P27 L613-616: Section title "Radar data traditional approaches" versus text "Although MPF optical-based retrieval outnumbers radar-based MPF retrievals, few MPF retrievals have been developed as well for radar sensors, such as SAR … followed by passive microwave…"– radar is an active instrument that emits the radiation it then receives, backscattered. Passive microwave sensors cannot be called radars. If you include PM sensors into the text, you need to include them also into the title as "Radar and Passive Microwave traditional approaches".

P30 L 700: "promising results when combining C-bad frequency" – C-band frequency

P33 L775-785 – I think the explanation of principal component analysis and k-means can be removed as it is widely known.

P 49 L1195-L1199: "For instance, although MODIS has a maximum latitude coverage of approximately 85 degrees, with some variations depending on the specific product or dataset being used, MODIS still can be used to generate sea ice data products for the entire Arctic region, including the North Pole. This is because MODIS is able to capture imagery of the Arctic region from various angles and perspectives, which can be used to create a composite view of the sea ice coverage." – there is no need to make composites of MODIS images or else generate something to cover the North pole. MODIS *viewing* angle reaches 65° at the swath edges as opposed to 0° in the middle of the swath, so that it covers the North Pole several times a day. The same is true for OLCI. Please correct.

P50 L1234: "the low resolution used for optical-based studies of meltdowns disregard smaller ponds" – first, melt ponds, not meltdowns. then, if not removed, this statement needs support, as spectral algorithms will still see the signature of even smaller ponds. For moderate resolving optical sensors, all ponds are subpixel anyway. Please clarify why small ponds would be omitted or for which exactly products.

P50 L1242 "Finally, (1) passive microwave sensor's brightness temperature is highly sensitive to atmospheric conditions" – this statement is wrong. it would be to a small extent true for 89GHz, but not for 6.9, 10 and 18GHz which take the main part in the manuscript. Please correct.